# MOSIC: MODEL-AGNOSTIC OPTIMAL SUBGROUP IDENTIFICATION WITH MULTI-CONSTRAINT FOR IMPROVED RELIABILITY

## ABSTRACT

Current subgroup identification methods typically follows a two-step approach: first estimate *conditional average treatment effects* (CATEs) and then apply thresholding or rule-based procedures to define subgroups. While intuitive, this decoupled approach fails to incorporate key constraints essential for real-world clinical decision-making—such as subgroup size and propensity overlap. These constraints operate on fundamentally different axes than CATE estimation and are not naturally accommodated within existing frameworks, thereby limiting the practical applicability of these methods. We propose a *unified optimization framework* that directly solves the *primal* constrained optimization problem to identify optimal subgroups. Our key innovation is a reformulation of the constrained primal problem as an *unconstrained differentiable min-max objective*, solved via a gradient descent-ascent algorithm. We theoretically establish that our solution converges to a feasible and locally optimal solution. Unlike threshold-based CATE methods that apply constraints as post-hoc filters, our approach enforces them directly during optimization. The framework is *model-agnostic*, compatible with a wide range of CATE estimators, and extensible to additional constraints like cost limits or fairness criteria. Extensive experiments on synthetic and real-world datasets demonstrate its effectiveness in identifying high-benefit subgroups while maintaining better satisfaction of constraints.

## 1 INTRODUCTION

In precision medicine, a fundamental challenge is identifying patient subgroups that benefit most from specific treatments, where heterogeneous effects must be estimated from observational data (Kosorok & Laber, 2019; Kravitz et al., 2004). Most existing methods adopt a two-step paradigm: they first estimate CATEs using machine learning methods, then deriving subgroups through either thresholding or simplified rule-based models (see Section 2).

However, this two-stage approach falls short in real-world settings, where subgroup identification must account for diverse, interacting constraints. Clinical deployment often requires satisfying statistical conditions like minimum subgroup size and overlap (VanderWeele et al., 2019; Crump et al., 2009), as well as operational and ethical considerations such as budget, safety, and fairness. Existing methods typically treat these constraints as post-hoc filters rather than integrating them into the optimization process. As a result, they *struggle to jointly satisfy multiple constraints*, leading to instability and poor performance. These challenges highlight a deeper disconnect between the continuous nature of CATE estimates and the discrete, constraint-driven structure of clinically actionable subgroup identification. This motivates the need for new frameworks that can incorporate and optimize over multiple real-world constraints in a unified and principled way.

We propose MOSIC (Model-agnostic Optimal Subgroup Identification with multi-Constraints), a *novel optimization framework* that identifies subgroups with maximal CATE while satisfying group size and overlap constraints—with flexibility to incorporate additional constraints. Our approach addresses the challenge of nonconvex/nonconcave optimization, and our contributions are threefold:

- **Problem Reformulation for Stable Solutions**: We develop a stable optimization procedure that: (1) formulates the task as a constrained problem (Section 3.1), (2) absorbs constraints into

the objective via a reformulation (Section 3.2), and (3) modify the objective to improve stability and solves it using a gradient descent-ascent algorithm(Section 4.1). Finally, we establish that the resulting solution is locally optimal and feasible (Section 4.2).

- **Flexibility**: MOSIC offers flexibility across three dimensions: (1) supporting multiple subgroup model architectures (e.g., multilayer perceptrons, decision trees) for different interpretability-performance tradeoffs, (2) compatibility with various ATE estimators, and (3) extensibility to diverse clinical constraints beyond our focus on size and overlap.
- **Comprehensive Evaluation**: We extensively evaluate our framework on both synthetic and real-world data, demonstrating its effectiveness in optimal subgroup identification under multiple constraints (Section 5). Our implementation is publicly available at `https://anonymous.4open.science/r/MOSIC3-8F13`.

## 2 RELATED WORK

We review treatment effect estimation, overlap handling, subgroup identification, and constrained optimization, highlighting that multi-constraints subgroup identification remains under-explored.

**Treatment Effect Estimation** MOSIC accommodates various average treatment effect (ATE) estimators. Traditional methods like IPTW, meta-learners (Künzel et al., 2019), R-learner (Nie & Wager, 2021), BART (Chipman et al., 2010) rely on either the treatment or outcome model, making them sensitive to model misspecification. In contrast, doubly robust estimators such as AIPTW (Robins et al., 1995), DR-learner (Kennedy, 2023), TMLE (Van Der Laan & Rubin, 2006), and DML (Chernozhukov et al., 2018) require only one model (treatment or outcome) to be correctly specified. This paper adopts AIPTW for illustration.

While ATE captures population-level effects, CATE estimation enables subgroup-specific recommendations. Modern methods include 1) tree-based methods like Causal Tree (CT) (Athey & Imbens, 2016), Causal Forest (CF) (Wager & Athey, 2018), and 2) neural-network-based approaches, such as TARNet (Shalit et al., 2017) and Dragonnet (Shi et al., 2019). In our AIPTW estimator, we estimate outcomes with Dragonnet and the treatment model with Logistic Regression (LR).

**Dealing with Limited Overlap** Limited sample overlap can bias treatment effect estimates or inflate variance. Common solutions include truncating propensity scores (Gruber et al., 2022; Cole & Hernán, 2008) and excluding low-overlap units (Crump et al., 2009; Schweisthal et al., 2024; Kallus, 2020; Li et al., 2018). We adopt the latter approach and incorporate a set of constraints to avoid low-overlap regions. This ensures more reliable ATE estimation within the identified subgroup.

**Optimal Subgroup Identification** Existing methods fall into three categories: (i) baseline methods without interpretability or constraints, (ii) *interpretable* methods, and (iii) *constrained* methods.

*Baseline methods* either: (1) rank patients by estimated CATE values (Cai et al., 2011; VanderWeele et al., 2019) (we benchmarked this approach employing CT, CF, and Dragonnet in Section 5) and DR-learner, R-learner, BART in Appendix E.2, or (2) optimize individual treatment rules using methods like Outcome-Weighted Learning (OWL) (Zhao et al., 2012; Liu et al., 2018). These methods provide useful benchmarks but do not readily accommodate additional constraints, a limitation our approach addresses.

*Interpretable methods* often rely on *Decision Tree* (DT) (Lipkovich et al., 2011; Dusseldorp et al., 2016; Huang et al., 2017; Athey & Wager, 2021). A representative example is Virtual Twins (VT) (Foster et al., 2011). It first estimates CATE and then applies a DT for interpretable subgroup identification. Beyond trees, rule learning approaches have been adopted Wang & Rudin (2021); Zhou et al. (2024). However, these methods rely on combinatorial searches and do not scale. Our method can instead leverage DTs as the backbone model, achieving the same level of interpretability while remaining scalable. We compare its performance against other DT-based methods in Section 5.3.

*Constrained methods* explicitly incorporate constraints into subgroup search. CAPITAL (Cai et al., 2022) is the most closely related approach to ours: it maximize subgroup size under a single constraint on subgroup ATE and allows extension to one additional constraint via Lagrangian relaxation. However, it struggles with multiple constraints due to instability and lack of feasibility guarantees. In Section 5, we compare our approach with VT, OWL, and CAPITAL.

Related ideas also appear in constrained policy learning, where the goal is to optimize policies subject to explicit safety and budget constraints. Examples include constrained policy optimiza-

tion (Achiam et al., 2017; Polosky et al., 2022), contextual bandits with knapsacks (Sivakumar et al., 2022; Badanidiyuru et al., 2018), and safe RL (García & Fernández, 2015; Zhang et al., 2025). These methods generally impose trajectory-level cumulative cost or risk constraints in sequential decision-making settings. In contrast, MOSIC addresses a different class of structural constraints aimed at improving the reliability of a learned subgroup rule in the static setting, such as the overlap and group-size constraints. It can additionally accommodate linear and ratio-form constraints such as risk and budget restrictions.

**Constrained Optimization** Constrained optimization algorithms vary by problem convexity. For *convex objective and constraints*, classical methods such as Projected Gradient Descent (PGD), Frank-Wolfe (FW), Interior Point Methods (IPM), and Lagrangian-based methods (e.g., Alternating Direction Method of Multipliers, ADMM (Boyd et al., 2011) are effective. For *non-convex objectives with convex feasible regions*, global optimality is NP-hard and convergence guarantees weaken (Lacoste-Julien, 2016; Wang et al., 2019). Our setting—*non-convex objective with non-convex constraints*—poses greater challenges: PGD struggles with complex projection, IPM scales poorly with constraint count, FW assumes convexity, and Lagrangian methods often lack stability and feasibility guarantees. ADMM fails here due to the non-separable objective. While ADMM variants exist for structured problems Gao et al. (2020), none apply to our setting. In contrast, our method guarantees both constraint feasibility and local optimality, critical for real-world deployment.

# 3 PROBLEM SETTINGS

Let $\boldsymbol{X} \in \mathcal{X} \subseteq \mathbb{R}^d$ denote baseline covariates, $A \in \mathcal{A} = \{0, 1\}$ the treatment (0: control, 1: treatment), and $Y \in \mathcal{Y}$ the outcome, which may be binary ($\mathcal{Y} = 0, 1$) or continuous ($\mathcal{Y} \subseteq \mathbb{R}$). The observational dataset consists of $n$ samples $(\boldsymbol{x}_i, a_i, y_i)_{i=1}^n$. Let $Y(0)$ and $Y(1)$ denote the potential outcomes under control and treatment. The propensity score is denoted as $e(\boldsymbol{x}) = P(A = 1 | \boldsymbol{X} = \boldsymbol{x})$, and potential outcomes as $\mu_a(\boldsymbol{x}) = \mathbb{E}[Y | \boldsymbol{X} = \boldsymbol{x}, A = a]$. We adopt standard causal inference assumptions: 1) Stable Unit Treatment Value Assumption (SUTVA): $Y = A \cdot Y(1) + (1 - A) \cdot Y(0)$; 2) Unconfoundedness: $\{Y(0), Y(1)\} \perp A | \boldsymbol{X}$; 3) Overlap: $0 < e(\boldsymbol{x}) < 1, \forall \boldsymbol{x} \in \mathcal{X}$.

## 3.1 RELIABLE SUBGROUP IDENTIFICATION FRAMEWORK

In this framework, our goal is to identify a patient subgroup with the largest ATE while ensuring reliability. We achieve this by learning a subgroup identification model $\tilde{S} : \mathbb{R}^d \mapsto \{0, 1\}$, which assigns a patient with covariates $\boldsymbol{X} = \boldsymbol{x}$ to the subgroup ($\tilde{S}(\boldsymbol{x}) = 1$) or excludes them ($\tilde{S}(\boldsymbol{x}) = 0$). We define the subgroup ATE as $\mathbb{E}[Y(1) - Y(0) | \tilde{S}(\boldsymbol{X}) = 1]$. Under SUTVA and unconfoundedness, this estimand can be expressed as $\mathbb{E}\left[\mathbb{E}[Y | A = 1, \boldsymbol{X}] - \mathbb{E}[Y | A = 0, \boldsymbol{X}] | \tilde{S}(\boldsymbol{X}) = 1\right]$.

Let $\hat{\phi}(\boldsymbol{x}_i, a_i, y_i)$ denote the estimated pseudo-outcomes for each sample, and $\mathbb{1}(\cdot)$ as the indicator function, a general estimator of the subgroup ATE is then

$$\frac{\sum_{i=1}^n \mathbb{1}(\tilde{S}(\boldsymbol{X}) = 1)\hat{\phi}(\boldsymbol{x}_i, a_i, y_i)}{\sum_{i=1}^n \mathbb{1}(\tilde{S}(\boldsymbol{X}) = 1)}.$$

In addition to maximizing the subgroup ATE, we introduce the following constraints:

- **Minimum size requirement.** A sufficiently large subgroup is essential for reliable estimation, robust statistical power, and economic viability in applications like drug repurposing.
- **Each sample in the identified subgroup has strong overlap.** Similar to excluding samples with extreme propensity scores (Crump et al., 2009), we impose that *each sample* in the selected subgroup has a propensity score bounded away from 0 and 1.

The above task can be formally stated as follows:

$$\min_{\tilde{S}} \ -\frac{\sum_{i=1}^n \mathbb{1}(\tilde{S}(\boldsymbol{X}) = 1)\hat{\phi}(\boldsymbol{x}_i, a_i, y_i)}{\sum_{i=1}^n \mathbb{1}(\tilde{S}(\boldsymbol{X}) = 1)} \qquad \textbf{(Problem I)}$$

$$\text{s.t.} \quad \mathbb{E}\left[\tilde{S}(\boldsymbol{X})\right] \geq c$$

$$\alpha \leq e(\boldsymbol{x}) \leq 1 - \alpha, \forall \boldsymbol{x} : \tilde{S}(\boldsymbol{x}) = 1, \qquad (1)$$

where $c \in (0, 1)$ is the desired subgroup size, and $\alpha \in [0, 0.5)$ is the threshold controlling the overlap constraint. Beyond the size and overlap, our method naturally accommodates more general **linear and ratio-form constraints**, which we formally introduce in Lemma 2 and Remark 2 (Section 4.2).

## 3.2 Relaxing the Combinatorial Formulation

Since the **Problem I** is combinatorial and difficult to optimize, we relax the subgroup identification to a probabilistic assignment. This relaxation is implemented using a parametric surrogate model $S : \mathbb{R}^d \mapsto (0, 1)$ with parameters $\boldsymbol{\theta}$. The ATE on the identified subgroup can be expressed as

$$f(\boldsymbol{\theta}) = \frac{\sum_{i=1}^n S(\boldsymbol{x}_i; \boldsymbol{\theta}) \hat{\phi}(\boldsymbol{x}_i, a_i, y_i)}{\sum_{i=1}^n S(\boldsymbol{x}_i; \boldsymbol{\theta})}. ^1 \tag{2}$$

Let $\hat{e}(\boldsymbol{x}_i)$, $\hat{\mu}_a(\boldsymbol{x}_i)$ denote the estimated $e(\boldsymbol{x}_i)$ and $\mu_a(\boldsymbol{x}_i)$. We then adopt the AIPTW estimator, which can be expressed as functions of $\hat{e}(\boldsymbol{x}_i)$ and $\hat{\mu}_a(\boldsymbol{x}_i)$:

$$\hat{\phi}_{\text{aiptw}}(\boldsymbol{x}_i, a_i, y_i) = \hat{\mu}_1(\boldsymbol{x}_i) - \hat{\mu}_0(\boldsymbol{x}_i) + \frac{a_i}{\hat{e}(\boldsymbol{x}_i)}(y_i - \hat{\mu}_1(\boldsymbol{x}_i)) - \frac{1 - a_i}{1 - \hat{e}(\boldsymbol{x}_i)}(y_i - \hat{\mu}_0(\boldsymbol{x}_i)).$$

While this study focus on AIPTW, $\hat{\phi}(\boldsymbol{x}_i, a_i, y_i)$ can be derived using other ATE estimators, making this formulation flexible. More details are illustrated in Appendix C.1.

Due to the relaxation of subgroup identification into a probabilistic assignment, the overlap constraint in equation 1, originally designed for discrete subgroup selection, must be adapted. To achieve this, we introduce $h(\boldsymbol{x}_i, \alpha)$, a surrogate function that reformulates the overlap constraint from **Problem I**:

$$h(\boldsymbol{x}_i, \alpha) = 1 - \frac{\hat{e}(\boldsymbol{x}_i)(1 - \hat{e}(\boldsymbol{x}_i))}{\alpha(1 - \alpha)}. \tag{3}$$

The following result, Lemma 1 (proof in Appendix B.1), establishes that the overlap constraint in **Problem I** can be replaced by a constraint on $h(\boldsymbol{x}_i, \alpha)$:

**Lemma 1.** $S(\boldsymbol{x}_i; \boldsymbol{\theta}) h(\boldsymbol{x}_i, \alpha) \leq 0$ *if and only if* $\alpha \leq \hat{e}(\boldsymbol{x}_i) \leq 1 - \alpha$.

With Lemma 1, our optimization can be reformulated as:

$$\min_{\boldsymbol{\theta}} \quad - f(\boldsymbol{\theta}) \qquad \qquad \textbf{(Problem II)}$$

$$\text{s.t.} \quad \frac{1}{n} \sum_{i=1}^n S(\boldsymbol{x}_i; \boldsymbol{\theta}) \geq c, \quad S(\boldsymbol{x}_i; \boldsymbol{\theta}) h(\boldsymbol{x}_i, \alpha) \leq 0, \ \forall i.$$

Solving **Problem II** remains a significant challenge. As shown in equation 2, the parametric model $S(\boldsymbol{x}_i; \boldsymbol{\theta})$ appears in both the numerator and denominator, making the objective function $f(\boldsymbol{\theta})$ neither convex nor concave, even for simple models like logistic regression. This nonconvexity also extends to the feasible set, rendering standard convex optimization methods, such as ADMM and FW, unsuitable. While Lagrangian relaxation could in principle be applied, doing so would require tuning a separate multiplier for each constraint, making the hyperparameter tuning process impractical at scale. To address this, we reformulate **Problem II** and present the final framework in Section 4.

## 4 Optimization Methods

Section 4.1 reformulates the task as a min-max optimization and adopts the $\gamma$-*Gradient Descent Ascent* ($\gamma$-GDA) algorithm (Schweisthal et al., 2024). Section 4.2 establishes feasibility guarantees, showing that MOSIC can identify the optimal subgroup while satisfying multiple constraints.

### 4.1 Min-Max Formulation and GDA

Since neither the objective nor the feasible region in **Problem II** is convex, we rewrite it using the saddle-point formulation:

$$L(\theta, \lambda) := -f(\boldsymbol{\theta}) + \boldsymbol{\lambda}^T \boldsymbol{g}(\boldsymbol{\theta}), \quad \min_{\boldsymbol{\theta}} \max_{\boldsymbol{\lambda} \geq 0} L(\theta, \lambda). \tag{4}$$

---

[1] $\hat{\phi}(\boldsymbol{x}_i, a_i, y_i)$ does not depend on parameter $\boldsymbol{\theta}$ as the estimation problem is separate from the parametric surrogate model $S$.

---

**Algorithm 1** $\gamma$-Gradient Descent Ascent ($\gamma$-GDA)

---

1: **Input:** step size $\eta$, decay rate $\zeta$, objective function $L(\boldsymbol{\theta}, \boldsymbol{\lambda})$.
2: Initialize $\boldsymbol{\lambda}_0 = 0$; Initialize $\boldsymbol{\theta}_0$ randomly
3: **for** $t = 0, 1, \ldots T$ **do**
4:     If converged, output $\boldsymbol{\theta}^* = \boldsymbol{\theta}_t$
5:     $\gamma \leftarrow (1 + t)^{\zeta}$
6:     Update $\boldsymbol{\theta}_t$ using gradient descent with learning rate $\eta/\gamma$: $\boldsymbol{\theta}_{t+1} \leftarrow \boldsymbol{\theta}_t - \left(\frac{\eta}{\gamma}\right) \nabla_{\boldsymbol{\theta}} L(\boldsymbol{\theta}_t, \lambda_t)$.
7:     Update $\boldsymbol{\lambda}_t$ using gradient ascent with learning rate $\eta$: $\boldsymbol{\lambda}_{t+1} \leftarrow \boldsymbol{\lambda}_t + \eta \nabla_{\boldsymbol{\lambda}} L(\boldsymbol{\theta}_{t+1}, \boldsymbol{\lambda}_t)$.
8: **end for**
9: **Output:** $\boldsymbol{\theta}^T$

---

We note that this is solving the primal constrained problem directly through a *min-max Lagrangian formulation*, unlike Lagrangian relaxation, which operates on the dual problem. We can solve this problem by $\gamma$-GDA (Jin et al., 2020), as described in Algorithm 1. The solution to **Problem III** is equivalent to solving **Problem II** (Boyd & Vandenberghe, 2004).

While the saddle-point formulation provides a correct representation of **Problem II**, applying standard GDA can lead to numerical instability and hinder convergence (See Appendix F for example). To obtain stable and convergent $\gamma$-GDA dynamics, additional structural conditions are needed. We thus refine the objective to satisfy two key properties:

- Only violated constraints contribute gradients. This can be achieved by introducing a ReLU transform on the constraint vector. When $g_i(\boldsymbol{\theta}) \leq 0$ (i.e., the constraint is satisfied), the penalty term becomes zero, so satisfied constraints no longer affect the optimization dynamics.
- Ensuring convergence to a feasible, locally optimal solution.

For the second requirement, we begin by defining local optimality in the context of a nonconvex-nonconcave min-max problem, introducing the concept of *local minmax point* (Definition 1). Informally, it is a fixed point where the objective remains stable under small perturbations in $\boldsymbol{\theta}$ (the parameters over which we minimize) and worst-case perturbations in $\boldsymbol{\lambda}$ (the parameters over which we maximize). Theorem 1 (Jin et al., 2020) states that if the min-max objective function is twice differentiable, then the $\gamma$-GDA algorithm, upon convergence, reaches either a local minmax point or a stationary point with a degenerate Hessian.

However, the Hessian of our objective in equation 4 is inherently degenerate: $\nabla^2_{\boldsymbol{\lambda}\boldsymbol{\lambda}} L(\boldsymbol{\theta}, \boldsymbol{\lambda}) = \mathbf{0}$, which hinders guarantees of local optimum convergence and feasibility. Unlike Nandwani et al. (2019), who did not exclude degenerate points–potentially invalidating their results–we introduce a modification that eliminates degenerate points, yielding the final objective of MOSIC:

$$L(\boldsymbol{\lambda}, \boldsymbol{\theta}) = -f(\boldsymbol{\theta}) + \boldsymbol{\lambda}^T ReLU(\boldsymbol{g}(\boldsymbol{\theta})) - \frac{\beta}{2}\boldsymbol{\lambda}^2, \quad \min_{\boldsymbol{\theta}} \max_{\boldsymbol{\lambda} \geq 0} L(\boldsymbol{\lambda}, \boldsymbol{\theta}), \qquad \textbf{(Problem III)}$$

where $\boldsymbol{\lambda} \in \mathbb{R}^{n+1}_+$, $\boldsymbol{g}(\boldsymbol{\theta}) = (S(\boldsymbol{x_1}; \boldsymbol{\theta})h(x_1; \alpha), \ldots, S(\boldsymbol{x_n}; \boldsymbol{\theta})h(\boldsymbol{x_n}; \alpha), c - \frac{1}{n}\sum_{i=1}^n S(\boldsymbol{x_i}; \boldsymbol{\theta})^\top$. For notational convenience, we write $\boldsymbol{g}(\boldsymbol{\theta})$ without explicitly indicating its dependence on fixed constraint values $c$ and $\alpha$ (or those for any additional constraint, if present).

### 4.2 FEASIBILITY GUARANTEES

With **Problem III**, we establish that if Algorithm 1 converges, it reaches to a *strict local minmax point* (Proof in Appendix B.2):

**Proposition 1** (Local Optimality). *Let $(\boldsymbol{\theta}', \boldsymbol{\lambda}')$ be a linearly stable point (formally defined in Definition 2) of Algorithm 1. Then, $(\boldsymbol{\theta}', \boldsymbol{\lambda}')$ must be a strict local minmax point.*

Building on this, the following result, Lemma 2, shows that upon convergence, all constraints are approximately satisfied within a small tolerance. This guarantee applies not only to the group size constraint but also to general linear constraints in $S$ (though not linear in $\boldsymbol{\theta}$), which include the overlap constraint.

**Lemma 2.** *Suppose the constraints include a group size constraint, $g^{size}(\boldsymbol{\theta}) = c - \frac{1}{n}\sum_{i=1}^n S(x_i; \boldsymbol{\theta})$, and $K$ ($K \geq 0$) additional constraints linear in $S$, $g^k(\boldsymbol{\theta}) = a^k + \sum_{i=1}^n b_i^k S(x_i; \boldsymbol{\theta})$, where*

---

$a^k, b_i^k \in \mathbb{R}$, and, $\forall k, |\sum_{i=1}^n b_i^k| > 0$. *Together, they define the constraint vector* $\boldsymbol{g}(\boldsymbol{\theta}) = (g^1(\boldsymbol{\theta}), \ldots, g^K(\boldsymbol{\theta}), g^{size}(\boldsymbol{\theta}))^\top$.

*Define* $\xi > 0$ *as the tolerance,* $\phi_{\max} = \max_i \hat{\phi}(x_i, a_i, y_i)$, $L = \sup |\partial S(\cdot; \boldsymbol{\theta})/\partial \theta_j|$ *as the coordinate-wise Lipschitz constant, and* $\mu_\Delta = \mathbb{E}[\partial S(x_i; \boldsymbol{\theta})/\partial \theta_j]$ *for* $j = \arg\max_j |\mu_\Delta|/L$.

*Let* $(\boldsymbol{\theta}^*, \boldsymbol{\lambda}^*)$ *be a strict local min-max point obtained by Algorithm 1. If*

$$\beta < \frac{\xi(c-\xi)|\mu_\Delta|}{2\,\phi_{\max}L},$$

*then either the model collapses (*$\frac{1}{n}\sum_{i=1}^n S(x_i; \boldsymbol{\theta}^*) < \xi$*) or all constraints are approximately satisfied:*

$$g^{size}(\boldsymbol{\theta}^*) \leq \xi, \quad g^k(\boldsymbol{\theta}^*) \leq \frac{\xi}{|\sum_{i=1}^n b_i^k|(1 + \frac{L}{|\mu_\Delta|\sqrt{n}}\sqrt{\log\frac{2}{\delta}})} \; w.p. \geq 1 - \delta$$

The proof of Lemma 2 (Appendix B.3) analyzes each constraint at *strict local minmax points* and establishes that, with $\beta$ properly chosen,[2] the constraints are either fully satisfied or violated up to a small tolerance error, if the model does not collapse. The proof further shows that, as long as the feasible region is non-negligible, the model is unlikely to collapse. [3]

**Remark 1** (Implication on the overlap constraint). *When* $n \to \infty$*, the bound on constraint violation is governed by* $|\sum_{i=1}^n b_i^k|$*. For the overlap constraint on sample* $j$*, this term reduces to* $h(x_j; \alpha)$ *since* $b_i^k = \mathbb{1}(i = j)h(x_j; \alpha)$*. When* $h(\boldsymbol{x_j}; \alpha)$ *is small, this denominator inflates the bound, suggesting potentially high violation. However, a small* $h(\boldsymbol{x_j}; \alpha)$ *indicates that the corresponding violation of the overlap condition is itself negligible. Thus, such constraints can be safely ignored in practice.*

**Remark 2** (Extension to ratio constraints). *Because the model maximizes the subgroup ATE, it naturally favors smaller subgroups by excluding samples with relatively low estimated CATE, while the group size constraint enforces a group size lower bound c. As a result, the group size typically converges to the size threshold c, i.e.,* $\sum_{i=1}^n S(x_i; \boldsymbol{\theta}) \approx c$ *at termination. This observation allows us to extend the linear constraint to ratio constraints*

$$g^k(\boldsymbol{\theta}) = a^k + \frac{\sum_{i=1}^n b_i^k S(\boldsymbol{x_i}; \boldsymbol{\theta})}{\sum_{i=1}^n S(\boldsymbol{x_i}; \boldsymbol{\theta})}.$$

*To retain compatibility with Lemma 2, we block gradient flow through the denominator during back-propagation, effectively treating it as a constant.*

The linear and ratio families capture many practical constraints relevant to healthcare applications. Linear constraints include the overlap constraint and budget constraints (each patient incurs a treatment cost under limited resources). Ratio constraints cover safety constraints (e.g., risk levels associated with patients) and certain fairness constraints (e.g. conditional statistical parity metric). Section 5.3 assesses MOSIC's extendability to these constraints. Due to the nonconvexity of both the objective and the feasible region, extending the guarantee to more complex constraints remains an open direction for future work.

Finally, we summarize our overall framework, MOSIC, in Algorithm 2. It first estimates the nuisances and computes the pseudo-outcomes for each sample, which are then fed into the objective function (**Problem III**). The optimization is subsequently performed using the $\gamma$-GDA algorithm.

## 5 EXPERIMENTS

We evaluate MOSIC on both synthetic and real-world data. Section 5.1 outlines the experimental setup. Section 5.2 demonstrate that MOSIC achieves high ATE while maintaining comparable covariate balance, or vice versa. Section 5.3 presents ablation studies, highlighting that MOSIC outperforms baselines under equal interpretability and extends to additional constraints.

---

[2]In practice, relaxing $\beta$ does not substantially increase constraint violation. $\beta \in [10^{-5}, 0.01]$ works well.

[3]In practice, if the model converges to a near-zero subgroup size, the run is restarted using a different random seed. Persistent collapse suggests that the feasible region defined by the constraints should be re-evaluated.

---

**Algorithm 2** MOSIC

---

1: **Input:** $\{(\boldsymbol{x_i}, a_i, y_i)\}_{i=1}^n$, constraint-related values $c$ and $\alpha$, learning rate $\eta$, decay rate $\zeta$
2: Estimate $\hat{\mu}_0(\boldsymbol{X})$, $\hat{\mu}_1(\boldsymbol{X})$, $\hat{e}(\boldsymbol{X})$ % *Estimate nuisance functions*
3: Compute $\hat{\phi}_{(}\boldsymbol{x}_i, a_i, y_i)$ using nuisance functions % *Pesudo-outcomes*
4: Construct $L(\boldsymbol{\theta}; \boldsymbol{\lambda}; c, \alpha)$
5: $\boldsymbol{\theta}^* \leftarrow \gamma\text{-GDA}(\eta, \zeta, L(\boldsymbol{\theta}; \boldsymbol{\lambda}; c, \alpha))$ % *Solve **Problem III** using Algorithm 1*
6: **Output:** Parametric surrogate model $S(\boldsymbol{X}; \boldsymbol{\theta}^*)$

---

## 5.1 SETUPS

**Datasets** We evaluate MOSIC on both synthetic and real-world datasets:

1. We generate synthetic data following a procedure adapted from Assaad et al. (2021) (details in Appendix C.2). We introduce an *imbalance parameter* $\tilde{\omega} \geq 0$ to determine the strength of confounding bias. In particular, we generated two datasets of size $n = 5,000$ with $d = 10$ covariates and the continuous outcome $Y$: (1) one with no confounding bias ($\tilde{\omega} = 0$) and (2) one with high confounding bias ($\tilde{\omega} = 5$).
2. We use two de-identified datasets from intensive care units (ICU): eICU (Pollard et al., 2018) and MIMIC-IV (Johnson et al., 2023). The eICU dataset ($n = 13,361$, $d = 23$ covariates) and the MIMIC-IV ($n = 6,516$, same covariates). In both datasets, treated patients ($A = 1$) received an initial Glucocorticoids dose of 160mg within 10 hours before to 24 hours after ICU admission. The outcome $Y$ represents 7-day survival, with $Y = 1$ indicating survival and $Y = 0$ otherwise. The covariates $\boldsymbol{X}$ include lab test results, vital signs, and sequential organ failure assessment (SOFA) scores (Vincent et al., 1996).

**Baselines** We compare MOSIC with two categories from Section 2: Those designed for subgroup identification and those adapted from CATE estimation algorithms.

1. Dedicated subgroup identification methods: We evaluate CAPITAL, OWL, and VT, modifying them to incorporate a group size constraint via thresholding (Details in Appendix C.3).These methods prioritize interpretability, making it unclear whether performance limits stem from this trade-off or poor CATE estimation. To address this, we consider the next category.
2. CATE estimation algorithms: We evaluate three methods: CT, CF, and Dragonnet in main results. We additionally compare with DR-learner, R-learner, BART, and an overlap-weighted variant of MOSIC (MOSIC-OW) in Appendix E.2. In these baselines, patients are ranked by the estimated CATE values and the top subgroup of the desired size is selected (VanderWeele et al., 2019). While not designed for subgroup selection, they provide a natural baseline. If CATE estimation were accurate, this approach would identify the optimal subgroup, allowing us to separate the impact of estimation reliability from the interpretability trade-off.

**Evaluation Metrics** We assess performance using two metrics: **(1) Subgroup ATE**, which measures whether identified subgroups achieve high ATE at the desired size. On synthetic data, we compute the ground-truth ATE; on real-world data, we use the difference between the subgroup and overall AIPTW estimates. **(2) ATE Estimation Reliability**. On synthetic data, it is measured by AIPTW estimation error. On real-world data, where the true ATE is unobserved, it is measured by the number of unbalanced features. A feature is considered unbalanced (Cohen, 2013) if its standardized mean difference (SMD) $> 0.2$ after IPTW reweighting (Austin, 2009; Zang et al., 2023). More unbalanced features indicate greater estimation uncertainty and lower subgroup reliability. To validate imbalance as a proxy for estimation error, we also report unbalanced features on synthetic data (Figure E.4.1). To assess how well MOSIC enforces the overlap constraint, we report the proportion of test-set samples that violate the overlap constraint on the real-world data.

**Implementation Details** For all datasets, we perform 100 random splits of the training and test sets. For each split, we first conduct a 5-fold cross-validation on the training set to determine the optimal hyperparameters (Appendix C.5). The model is then retrained on the entire training set using the selected hyperparameters and evaluated on the corresponding test set. The final results are reported using the mean and standard deviations across all 100 evaluations.

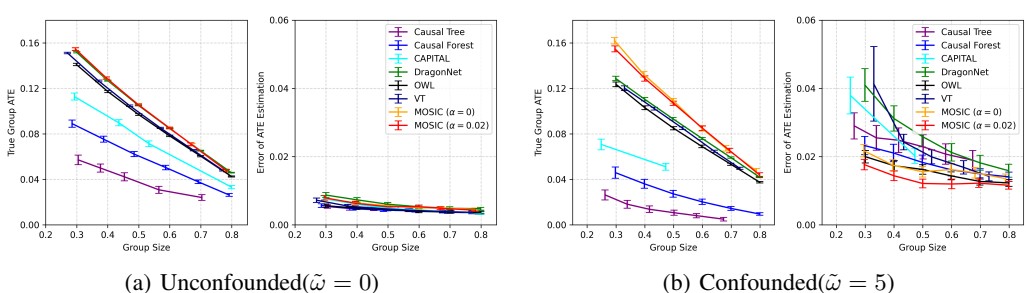

Figure 1: True ATE and estimation error across different group sizes on synthetic data

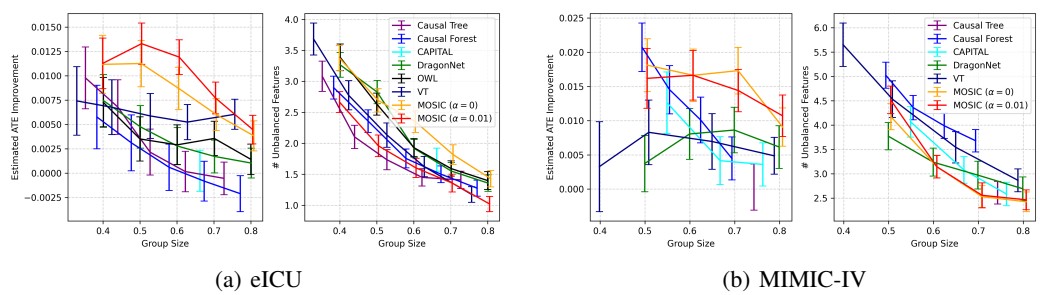

Figure 2: Estimated ATE and the number of unbalanced features on real-world datasets.[5]

We implement the subgroup identification model ($S$) using MLP and DT. In the next sections, 'MOSIC' refers to MOSIC-MLP, unless stated otherwise. For DTs, we adopt the neural-network representation of Marton et al. (2024). All these models are trained using Algorithm 1, with L1 regularization applied in the loss function. For nuisance function estimation, we use LR for the propensity score model $\hat{e}(\cdot)$, and Dragonnet for the outcome models $\hat{\mu}_a(\cdot)$ (Appendix C.4). [4] Our implementation shares data for nuisance estimation and subgroup selection. Because the test set is never used for either step, the final comparison on the test set is not affected by data sharing in evaluation. We also assess a sample-splitting variant in Appendix E.3.

### 5.2 RESULTS

**Synthetic Data** To assess the impact of overlap constraints, we compare MOSIC with ($\alpha = 0.02$) and without ($\alpha = 0$) them. Figure 1 demonstrates that MOSIC consistently identifies subgroups with the highest true subgroup ATE across all group sizes (Figure 1(a) and 1(b), left); and it achieves the lowest ATE estimation errors (Figure 1(a) and 1(b), right).

Further, we numerically verify that MOSIC can indeed satisfy the overlap constraint and investigate the correlation between the estimation error and the number of unbalanced features (Figure E.4.1). We also investigate the statistical properties of the proposed procedure in Appendix G and H.

**Real-World Data** Since the real-world datasets contain a large portion of samples with propensities outside [0.05,0.95] (Figure E.6.2), we relax the overlap constraint threshold to be $\alpha = 0.01$. Figures 2 and Figure E.2.1 demonstrate that MOSIC consistently outperforms other methods. We highlight the importance of jointly evaluating subgroup ATE and covariate balance when assessing performance. A large ATE alone is insufficient if covariate imbalance persists, as it may indicate unreliable estimates. As shown in Figures 2(a) and 2(b) (left), MOSIC achieves higher subgroup ATEs at comparable group sizes in most cases. Even when the ATE advantage is not statistically significant (e.g., MOSIC ($\alpha = 0.01$) vs. CF at $c = 0.6$ on MIMIC, p = 0.68), MOSIC delivers significantly better covariate balance (p = 0.000052; Figures 2(a) and 2(b), right). Full statistical

---

[4]Empirically, estimating propensity scores with Dragonnet results in a large number of unbalanced features; we therefore adopt LR, demonstrating the flexibility of MOSIC.

[5]The OWL method failed to converge on the MIMIC-IV dataset and was therefore omitted from the figure.

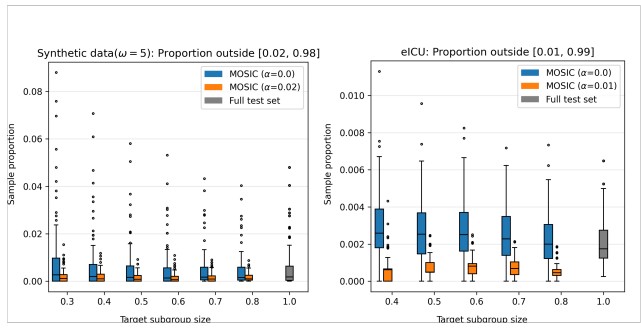

Figure 3: Overlap Evaluation on Test Set

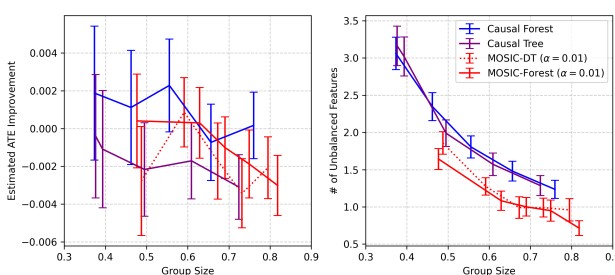

Figure 4: Results on eICU: MOSIC with DT backbone vs. DT-based baselines.

test results are provided in Appendix D. For completeness, feature imbalance with SMD> 0.1 as the threshold is also presented (Figure E.6.1).

Figure 3 (right) shows that MOSIC with the overlap constraint successfully limits the number of test-set samples falling outside the allowable propensity range, demonstrating effective enforcement of the constraint. This reduction in extreme-propensity samples leads to improved feature balance on eICU (Figures 2(a), right). To quantify uncertainty of the subgroup ATE, we additionally compute 95% confidence intervals for the subgroup ATEs and conduct sensitivity analysis of unmeasured confounding (Appendix E.1). For completeness, we also report training curves showing stable optimization (Figure E.9.1).

### 5.3 ABLATION STUDIES

**Decision Tree as Backbone** While MOSIC-MLP outperforms baselines, its black-box nature limits interpretability. In contrast, decision trees are favored in clinical settings for their transparency (Cai et al., 2022). We therefore compare MOSIC with DT backbone to DT-based baselines (CT, CF) on eICU. VT is excluded due to its high ATE estimation error on synthetic data.(Figure 1(b)).

To impose a fair comparison, we match model capacity: we fix the tree depth to 5 for both MOSIC (denoted as MOSIC-DT) and CT, and use ensembles of 3 trees of depth 5 for MOSIC (denoted as MOSIC-Forest) and CF. As shown in Figure 4, MOSIC consistently outperforms CT and CF under the same interpretability requirement. Notably, despite the limited model capacity, MOSIC effectively enforces both group size and overlap constraints, leading to improved covariate balance.

**Extension to Additional Constraints** MOSIC readily extends to other constraints. On synthetic data, we evaluated its performance when additional requirements were imposed on top of the size and overlap constraint: first adding a safety constraint, then safety and budget constraints, and finally safety, budget, and fairness constraints. MOSIC can effectively satisfy all of them (Appendix E.7).

On eICU, in addition to the size ($c = 0.4$) and overlap constraint ($\alpha = 0.01$), we introduced a safety constraint motivated by evidence that glucocorticoids may exacerbate neural damage (Hill & Spencer-Segal, 2021). In particular, we required that the proportion of patients with a Glasgow Coma Scale (GCS) score $< 6$, the most severe level of neural dysfunction in SOFA, remain below 0.05. Table 1 shows that MOSIC can additionally satisfy this safety constraint (See Appendix E.8 for details). The DT example in Figure 5 shows that the constraint introduces an explicit rule excluding

Table 1: Results on eICU with the additional constraint requiring the proportion of patients with GCS< 6 to remain below 0.05. Reported values are mean $\pm$ standard error over 100 random splits.

| Metric | Constraint: Size & Overlap | Constraint: Size & Overlap & GCS |
|---|---|---|
| Group Size | $0.46 \pm 0.10$ | $0.44 \pm 0.10$ |
| # Unbalanced Features | $2.0 \pm 1.72$ | $2.2 \pm 2.28$ |
| ATE Improvement | $0.02 \pm 0.02$ | $0.02 \pm 0.02$ |
| Proportion of GCS< 6 | $\mathbf{0.15 \pm 0.05}$ | $\mathbf{0.03 \pm 0.03}$ |

low-GCS patients. Because DTs are sensitive to randomness, we also run MOSIC-MLP and conduct SHAP analyses. Figure E.8.1 shows that GCS is weakly used before adding the constraint, but becomes strongly aligned with it afterward, confirming consistent behavior across models.

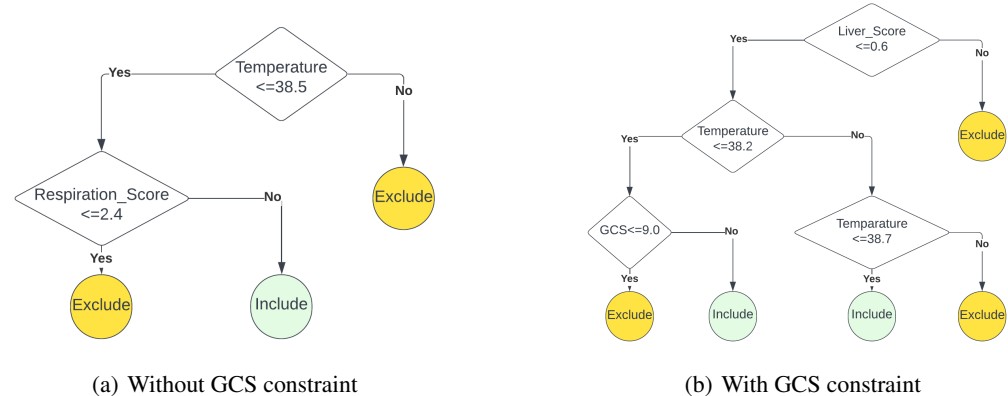

(a) Without GCS constraint          (b) With GCS constraint

Figure 5: A returning DT from the same split on eICU with and without the GCS< 6 constraint.

**Runtime Analysis and Training Dynamics**    Gradient-based methods like GDA scale well in high-dimensional settings because each update only computes gradients, giving a per-iteration cost linear in the number of parameters. This makes our method more scalable than combinatorial methods such as CAPITAL. Each constraint adds one evaluation per update. For the linear and ratio-form constraints considered in this work, evaluating a constraint requires at most $O(n)$ operations, where $n$ is the sample size. Thus, computing the loss has per-iteration cost $O(nm)$ for $m$ constraints. Although our formulation includes $n$ overlap constraints, each one contributes only a single multiplication, making the total overlap penalty cost $O(n)$, not $O(n^2)$. Consequently, the overall per-iteration cost remains low despite the large number of constraints in our problem.

We also provide empirical runtime analysis for all methods. Experiments are run on CPU to mirror resource-limited clinical environments. Table I.0.1 shows that this overhead is small relative to nuisance estimation, indicating that computation is unlikely to be a deployment bottleneck.

## 6   Conclusion

We propose MOSIC, a model-agnostic framework for optimal subgroup identification that handles multiple constraints with feasibility guarantees. It demonstrates strong empirical performance under group size and overlap constraints, flexibly extends to additional clinical constraints, and supports diverse models to deliver interpretable solutions.

Finally, we acknowledge that the real-world ICU datasets used in this study have known limitations, including potential immortal-time bias and unmeasured confounding. Therefore, subgroup rules obtained from these datasets (e.g., Appendix E.8) should be interpreted as illustrative rather than clinical guidance. These intrinsic limitations affect all subgroup identification and CATE-based approaches equally, and does not affect the conclusion of our comparison. In addition, our ablation study using a GCS-based safety constraint demonstrates how domain knowledge can be integrated to refine subgroup definitions when needed.

## 7 REPRODUCIBILITY STATEMENT

We have taken several steps to ensure the reproducibility of our results. The theoretical foundations of MOSIC, including assumptions, proofs of feasibility and local optimality, and supporting lemmas, are provided in the Appendix A and B. The optimization framework and algorithms are described in detail in Sections 3 and 4, with full pseudocode in Algorithm 1 and 2. Experimental protocols, including dataset, implementation details, evaluation metrics, and hyperparameter tuning, are documented in Section 5 and Appendix C. We evaluate our method on both synthetic and real-world ICU datasets, with synthetic data generation procedures detailed in Appendix C.2, and we report results over 100 random splits to assess robustness. To facilitate replication, we provide an anonymous implementation at https://anonymous.4open.science/r/MOSIC3-8F13 .

## 8 THE USE OF LARGE LANGUAGE MODELS (LLMS)

LLMs were employed to polish the writing of this manuscript, assist in identifying related work, and draft the README documentation for the accompanying code repository.

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

# A  PRELIMINARY DEFINITIONS AND THEOREMS

## A.1  LOCAL MINMAX POINT

**Definition 1** (Local minmax point). *A point $(\theta^\star, \lambda^\star)$ is said to be a local minmax point of $L$, if there exists $\delta_0 > 0$ and a continuous function $h$ satisfying $h(\delta) \to 0$ as $\delta \to 0$, such that for any $\delta \leq \delta_0$, and any $(\theta, \lambda)$ satisfying*

$$\|\theta - \theta^\star\| \leq \delta \quad and \quad \|\lambda - \lambda^\star\| \leq h(\delta),$$

*we have*

$$L(\theta^\star, \lambda) \leq L(\theta^\star, \lambda^\star) \leq \max_{\lambda': \|\lambda' - \lambda^\star\| \leq h(\delta)} L(\theta, \lambda').$$

*If a point $(\theta^\star, \lambda^\star)$ satisfy*

$$[\nabla^2_{\theta\theta} L - \nabla^2_{\theta\lambda} L (\nabla^2_{\lambda\lambda} L)^{-1} \nabla^2_{\lambda\theta} L] \succ 0,$$

*we call it a strict local minmax point Jin et al. (2020).*

## A.2  LINEARLY STABLE POINT

**Definition 2** (Linearly Stable Point). *For a differentiable dynamical system $\mathbf{w}$, a fixed point $\mathbf{z}^\star$ is a linearly stable point of $\mathbf{w}$ if its Jacobian matrix $\mathbf{J}(\mathbf{z}^\star) := \left(\frac{\partial \mathbf{w}}{\partial \mathbf{z}}\right)(\mathbf{z}^\star)$ has spectral radius $\rho(\mathbf{J}(\mathbf{z}^\star)) \leq 1$.*

## A.3 CONVERGENCE OF $\gamma$-GDALGORITHM

**Theorem 1.** *(Jin et al Jin et al. (2020), Theorem 26): Given an objective:* $\min\limits_{x\ in\mathcal{X}} \max\limits_{y\in\mathcal{Y}} f(x,y)$*. For any twice-differentiable f, the strict linearly stable limit points of the $\gamma$-GDA flow are {strict local minmax points} $\cup$ {$(\theta,\lambda)$ — $(\theta,\lambda)$ is stationary and $\nabla^2_{\lambda\lambda} f(\theta,\lambda)$ is degenerate} as $\gamma \to \infty$.*

## B PROOFS

### B.1 PROOF OF LEMMA 1

**Lemma 1.** $S(\boldsymbol{x}_i;\boldsymbol{\theta})h(\boldsymbol{x}_i,\alpha) \leq 0$ *if and only if* $\alpha \leq \hat{e}(\boldsymbol{x}_i) \leq 1 - \alpha$.

*Proof.* We proceed by proving both directions separately.

**Forward Direction:** Suppose $S(x_i;\theta)h(x_i) \leq 0$. We aim to show that this implies $\alpha \leq e(x_i) \leq 1 - \alpha$.

$$
\begin{aligned}
&S(x_i;\theta)h(x_i) \leq 0 \\
&\implies h(x_i) \leq 0 \qquad \text{(Since } S(x_i) > 0) \\
&\implies 1 - \frac{e(x_i)(1 - e(x_i))}{\alpha(1 - \alpha)} \leq 0 \qquad \text{(Substituting } h(x_i)) \\
&\implies \alpha(1 - \alpha) - e(x_i)(1 - e(x_i)) \leq 0 \\
&\implies e(x_i) - \alpha - \left(e(x_i)^2 - \alpha^2\right) \geq 0 \\
&\implies (e(x_i) - \alpha)(1 - (e(x_i) + \alpha)) \geq 0 \\
&\implies (e(x_i) - \alpha)(1 - \alpha - e(x_i)) \geq 0 \\
&\implies \alpha \leq e(x_i) \leq 1 - \alpha.
\end{aligned}
$$

**Backward Direction:** Suppose $\alpha \leq e(x_i) \leq 1 - \alpha$. We need to show that this implies $S(x_i;\theta)h(x_i) \leq 0$.

Define the auxiliary function $q(z) = z(1 - z)$, whose derivative is given by:

$$q'(z) = 1 - 2z.$$

Thus, $q(z)$ attains its maximum at $z = 0.5$.

When $0 < \alpha \leq e(x_i) \leq 0.5$, we have $q'(e(x_i)) = 1 - 2e(x_i) \geq 0$, which implies that $q(e(x_i))$ is non-decreasing on $[0, 0.5]$. Consequently, from the assumption $\alpha \leq e(x_i)$, we obtain:

$$q(e(x_i)) \geq q(\alpha) \implies e(x_i)(1 - e(x_i)) \geq \alpha(1 - \alpha).$$

- Similarly, for $0.5 < e(x_i) \leq 1 - \alpha < 1$, we have $e(x_i)(1 - e(x_i)) \geq \alpha(1 - \alpha)$.

By combining both cases, we conclude that:

$$e(x_i)(1 - e(x_i)) \geq \alpha(1 - \alpha).$$

Dividing both sides by $\alpha(1 - \alpha)$ yields:

$$1 - \frac{e(x_i)(1 - e(x_i))}{\alpha(1 - \alpha)} = h(x_i) \leq 0.$$

Since $S(x_i) > 0$, it follows that:

$$S(x_i)h(x_i) \leq 0.$$

This completes the proof. $\qquad\square$

### B.2 PROOF OF PROPOSITION 1

**Proposition 1** (Local Optimality). *Let $(\boldsymbol{\theta}', \boldsymbol{\lambda}')$ be a linearly stable point (formally defined in Definition 2) of Algorithm 1. Then, $(\boldsymbol{\theta}', \boldsymbol{\lambda}')$ must be a strict local minmax point.*

*Proof.* The derivation of Theorem 1 relies on the Jacobian matrix and requires twice differentiability. However, our objective function incorporates ReLU activations, introducing non-differentiability at the origin. This prevents us from directly applying Theorem 1 in its standard form. Nonetheless, since the probability of encountering exact zero inputs to ReLU is negligible, our objective function remains effectively differentiable in practice when using gradient-based optimization.

Thus, the key arguments of Theorem 1 extend to our objective (**Problem III**), implying that its stable limit points must either be local minmax points or points where the second derivative is degenerate.

Further, we compute the first and second derivatives of $L$ with respect to $\lambda$:

$$\frac{\partial L}{\partial \lambda_i} = \text{ReLU}(g_i(\theta)) - \beta\lambda_i, \tag{5}$$

$$\frac{\partial^2 L}{\partial \lambda_i \lambda_j} = \begin{cases} 0 & \text{if } i \neq j, \\ -\beta & \text{if } i = j. \end{cases} \tag{6}$$

Since $\beta > 0$, the Hessian matrix $\frac{\partial^2 L}{\partial \lambda_i \lambda_j}$ is never degenerate. Applying Theorem 2, we conclude that the strict linearly stable limit points of the $\gamma$-GDA flow are precisely the set of local minmax points. $\qquad\square$

### B.3 PROOF OF LEMMA 2

**Lemma 2.** *Suppose the constraints include a group size constraint, $g^{size}(\boldsymbol{\theta}) = c - \frac{1}{n}\sum_{i=1}^{n} S(x_i; \boldsymbol{\theta})$, and $K$ ($K \geq 0$) additional constraints linear in $S$, $g^k(\boldsymbol{\theta}) = a^k + \sum_{i=1}^{n} b_i^k S(x_i; \boldsymbol{\theta})$, where $a^k, b_i^k \in \mathbb{R}$, and, $\forall k, |\sum_{i=1}^{n} b_i^k| > 0$. Together, they define the constraint vector $\boldsymbol{g}(\boldsymbol{\theta}) = (g^1(\boldsymbol{\theta}), \ldots, g^K(\boldsymbol{\theta}), g^{size}(\boldsymbol{\theta}))^\top$.*

*Define $\xi > 0$ as the tolerance, $\phi_{\max} = \max_i \hat{\phi}(x_i, a_i, y_i)$, $L = \sup |\partial S(\cdot; \boldsymbol{\theta})/\partial\theta_j|$ as the coordinate-wise Lipschitz constant, and $\mu_\Delta = \mathbb{E}[\partial S(x_i; \boldsymbol{\theta})/\partial\theta_j]$ for $j = \arg\max_j |\mu_\Delta|/L$.*

*Let $(\boldsymbol{\theta}^*, \boldsymbol{\lambda}^*)$ be a strict local min-max point obtained by Algorithm 1. If*

$$\beta < \frac{\xi(c-\xi)|\mu_\Delta|}{2\,\phi_{\max}L},$$

*then either the model collapses ($\frac{1}{n}\sum_{i=1}^{n} S(x_i; \boldsymbol{\theta}^*) < \xi$) or all constraints are approximately satisfied:*

$$g^{size}(\boldsymbol{\theta}^*) \leq \xi, \quad g^k(\boldsymbol{\theta}^*) \leq \frac{\xi}{|\sum_{i=1}^{n} b_i^k|(1 + \frac{L}{|\mu_\Delta|\sqrt{n}}\sqrt{\log\frac{2}{\delta}})} \; w.p. \geq 1 - \delta$$

*Proof.* At convergence, the following condition holds:

$$\frac{\partial L}{\partial \boldsymbol{\lambda}}\Big|_{\boldsymbol{\lambda}=\boldsymbol{\lambda}^*} = ReLU(\boldsymbol{g}(\boldsymbol{\theta}^*)) - \beta\boldsymbol{\lambda}^* = \mathbf{0}, \tag{7}$$

$$\frac{\partial L}{\partial \boldsymbol{\theta}}\Big|_{\boldsymbol{\theta}=\boldsymbol{\theta}^*} = -\frac{\partial f}{\partial \boldsymbol{\theta}}\Big|_{\boldsymbol{\theta}=\boldsymbol{\theta}^*} + \boldsymbol{\lambda}^*\frac{\partial ReLU}{\partial \boldsymbol{g}} \cdot \frac{\partial \boldsymbol{g}}{\partial \boldsymbol{\theta}}\Big|_{\boldsymbol{\theta}=\boldsymbol{\theta}^*} = \mathbf{0} \tag{8}$$

Since Eq. equation 7 applied component-wise to $\boldsymbol{g}(\boldsymbol{\theta}^*)$, we analyze each individual component $g(\boldsymbol{\theta}^*)$ separately. Due to ReLU, we distinguish between two cases: $g(\boldsymbol{\theta}^*) \leq 0$ and $g(\boldsymbol{\theta}^*) > 0$. When $g(\boldsymbol{\theta}^*) \leq 0$, Lemma 2 holds trivially. Therefore, we focus on the remaining case where $g(\boldsymbol{\theta}^*) > 0$, which means the constraint is violated and Eq. equation 7 and Eq.equation 8 simplifies to

$$g(\boldsymbol{\theta}^*) = \beta\lambda^*, \tag{9}$$

$$-\frac{\partial f}{\partial \boldsymbol{\theta}}\Big|_{\boldsymbol{\theta}=\boldsymbol{\theta}^*} + \lambda^* \frac{\partial g}{\partial \boldsymbol{\theta}}\Big|_{\boldsymbol{\theta}=\boldsymbol{\theta}^*} = \mathbf{0} \tag{10}$$

Substituting Eq.equation 9 into Eq.equation 10, we obtain

$$-\frac{\partial f}{\partial \boldsymbol{\theta}}\Big|_{\boldsymbol{\theta}=\boldsymbol{\theta}^*} + \frac{g(\boldsymbol{\theta}^*)}{\beta} \frac{\partial g}{\partial \boldsymbol{\theta}}\Big|_{\boldsymbol{\theta}=\boldsymbol{\theta}^*} = 0. \tag{11}$$

We note that $g(\boldsymbol{\theta}^*)$ is a scalar, meaning that each elements of $\frac{\partial f}{\partial \boldsymbol{\theta}}\big|_{\boldsymbol{\theta}=\boldsymbol{\theta}^*}$ is proportional to the corresponding elements in $\frac{\partial g}{\partial \boldsymbol{\theta}}\big|_{\boldsymbol{\theta}=\boldsymbol{\theta}^*}$ by the same constant $\frac{g(\boldsymbol{\theta}^*)}{\beta} > 0$. This allows us to pick any element of $\boldsymbol{\theta}$ to analyze $g(\boldsymbol{\theta}^*)$. (Note that when $\theta \in \mathbb{R}^d$ is high-dimensional, Eq.equation 11 is challenging to achieve, because we need one scalar to satisfy all $d$ equations. In practice, we observe small oscillations when the algorithm converges, which implies the algorithm finds it hard to exactly satisfy all equations.)

Denote $\phi_i = \phi(\boldsymbol{x}_i, a_i, y_i)$, $\Delta_i = \frac{\partial S(\boldsymbol{x}_i; \boldsymbol{\theta})}{\partial \theta_j}\big|_{\boldsymbol{\theta}=\boldsymbol{\theta}^*} \leq L$, $\mu_\Delta = \mathbb{E}\Delta_i$, where $j = \arg\max_j \frac{\mu_\Delta}{L}$. The definition of function $f$ (Eq.equation 2) results in

$$\left|\frac{\partial f}{\partial \theta_j}\Big|_{\boldsymbol{\theta}=\boldsymbol{\theta}^*}\right| = \left|\frac{\sum_{i=1}^n \phi_i \Delta_i \sum_{s=1}^n S(\boldsymbol{x_s}; \boldsymbol{\theta})}{\left(\sum_{i=1}^n S(\boldsymbol{x}_i; \boldsymbol{\theta})\right)^2} - \right.$$

$$\left.\frac{\sum_{i=1}^n \Delta_i \sum_{s=1}^n \phi_s S(\boldsymbol{x_s}; \boldsymbol{\theta})}{\left(\sum_{i=1}^n S(\boldsymbol{x}_i; \boldsymbol{\theta})\right)^2}\right|$$

$$= \left|\frac{\sum_{i=1}^n \sum_{s=1}^n \Delta_i S(\boldsymbol{x_s}; \boldsymbol{\theta})(\phi_i - \phi_s)}{\left(\sum_{i=1}^n S(\boldsymbol{x}_i; \boldsymbol{\theta})\right)^2}\right|$$

With $\phi_{max} = \max_i |\hat{\phi}(x_i, a_i, y_i)|$, we have,

$$\left|\frac{\partial f}{\partial \theta_j}\Big|_{\boldsymbol{\theta}=\boldsymbol{\theta}^*}\right| \leq 2\phi_{max}\left|\frac{\sum_{i=1}^n \sum_{s=1}^n \Delta_i S(\boldsymbol{x_s}; \boldsymbol{\theta})}{\left(\sum_{i=1}^n S(\boldsymbol{x}_i; \boldsymbol{\theta})\right)^2}\right|$$

$$= 2\phi_{max}\left|\frac{\sum_{i=1}^n \Delta_i \sum_{s=1}^n S(\boldsymbol{x_s}; \boldsymbol{\theta})}{\left(\sum_{i=1}^n S(\boldsymbol{x}_i; \boldsymbol{\theta})\right)^2}\right|$$

$$= 2\phi_{max}\frac{\left|\sum_{i=1}^n \Delta_i\right|}{\sum_{i=1}^n S(\boldsymbol{x}_i; \boldsymbol{\theta})} \tag{12}$$

Substituting Eq. equation 12 into Eq. equation 11 and take assoluate value, we obtain

$$g(\boldsymbol{\theta}^*) \leq \beta \left(\left|\frac{\partial g}{\partial \boldsymbol{\theta}}\Big|_{\boldsymbol{\theta}=\boldsymbol{\theta}^*}\right|\right)^{-1} \frac{2\phi_{max}|\sum_{i=1}^n \Delta_i|}{\sum_{i=1}^n S(\boldsymbol{x}_i; \boldsymbol{\theta})} \tag{13}$$

Next, we will prove the upper bound of the group size constraint violation and the general linear constraints violation by analyzing $\frac{\partial g}{\partial \boldsymbol{\theta}}\big|_{\boldsymbol{\theta}=\boldsymbol{\theta}^*}$.

**Case 1:** $g(\boldsymbol{\theta}^*) = c - \frac{1}{n}\sum_{i=1}^n S(\boldsymbol{x}_i; \boldsymbol{\theta}^*) > 0$: the group size constraint is violated.

$$\frac{\partial g}{\partial \theta_j}\Big|_{\boldsymbol{\theta}=\boldsymbol{\theta}^*} = -\frac{1}{n}\sum_{i=1}^n \Delta_i$$

Substituting this into Eq. equation 13:

$$g(\boldsymbol{\theta}^*) \leq \beta \left(\left|-\frac{1}{n}\sum_{i=1}^n \Delta_i\right|\right)^{-1} \frac{2\phi_{max}|\sum_{i=1}^n \Delta_i|}{\sum_{i=1}^n S(\boldsymbol{x}_i; \boldsymbol{\theta})}$$

$$= \beta \frac{2\phi_{max}}{\frac{1}{n}\sum_{i=1}^n S(\boldsymbol{x}_i; \boldsymbol{\theta})} \tag{14}$$

Substitute $\beta \leq \frac{\xi(c-\xi)\mu_\Delta}{2\phi_{max}L}$ into Eq. equation 14, we obtain

$$g(\boldsymbol{\theta^*}) \leq \frac{\xi(c-\xi)\mu_\Delta}{\frac{1}{n}\sum_{i=1}^n S(\boldsymbol{x}_i; \boldsymbol{\theta})L} \leq \frac{\xi(c-\xi)}{\frac{1}{n}\sum_{i=1}^n S(\boldsymbol{x}_i; \boldsymbol{\theta})} = \frac{\xi(c-\xi)}{c-g(\boldsymbol{\theta^*})}$$

Solving for $g(\boldsymbol{\theta^*})$, we obtain $g(\boldsymbol{\theta^*}) \leq \xi$ (group size constraint satisfied) or $g(\boldsymbol{\theta^*}) \geq c - \xi$ (the model collapses, group size is near zero).

Next, we show that the model is improbable to collapse if the feasible region is non-negligible.

Define the collapsed region as

$$\boldsymbol{\Theta}_{\text{collapse}} := \left\{ \boldsymbol{\theta} \in \Theta : \frac{1}{n}\sum_{i=1}^n S(\boldsymbol{x}_i; \boldsymbol{\theta}) < \xi \right\},$$

and the feasible region as

$$\boldsymbol{\Theta}_{\text{feasible}} := \left\{ \boldsymbol{\theta} \in \Theta : \frac{1}{n}\sum_{i=1}^n S(\boldsymbol{x}_i; \boldsymbol{\theta}) \geq c \right.$$

$$\left. \text{and other constraints are met} \right\}$$

for some thresholds $0 < \xi \ll c < 1$. Suppose that the volume of the feasible region dominates that of the collapsed region, i.e.,

$$|\boldsymbol{\Theta}_{\text{collapse}}| \ll |\boldsymbol{\Theta}_{\text{feasible}}|.$$

Given $\forall i, 0 \leq S(\boldsymbol{x}_i; \boldsymbol{\theta}) \leq 1$, using Hoeffding inequality, we obtain

$$P\left( \sum_{i=1}^n S(\boldsymbol{x}_i; \boldsymbol{\theta}) - \mathbb{E}\sum_{i=1}^n S(\boldsymbol{x}_i; \boldsymbol{\theta}) \leq -t \right) \leq \exp(-\frac{2t^2}{n}).$$

Let $t = \sqrt{\frac{n\log(1/\delta)}{2}}$, with probability at least $1 - \delta$, we have

$$\sum_{i=1}^n S(\boldsymbol{x}_i; \boldsymbol{\theta}) - \mathbb{E}\sum_{i=1}^n S(\boldsymbol{x}_i; \boldsymbol{\theta}) \geq -\sqrt{\frac{n\log(1/\delta)}{2}}$$

Assume $\boldsymbol{\theta}$ is sampled uniformly from $\Theta_{\text{collapase}} \cup \Theta_{\text{feasible}}$, $|\Theta_{collapase}| \ll |\Theta_{feasible}| \Rightarrow P(\boldsymbol{\theta} \in \Theta_{\text{collapse}}) \ll P(\boldsymbol{\theta} \in \Theta_{\text{feasible}})$. Then,

$$\mathbb{E}\sum_{i=1}^n S(\boldsymbol{x}_i; \boldsymbol{\theta}) = P(\boldsymbol{\theta} \in \Theta_{collapse}) \cdot \sum_{i=1}^n S(\boldsymbol{x}_i; \boldsymbol{\theta})$$

$$+ P(\boldsymbol{\theta} \in \Theta_{feasible}) \cdot \sum_{i=1}^n S(\boldsymbol{x}_i; \boldsymbol{\theta})$$

Given that the contribution from $\Theta_{\text{collapse}}$ is negligible, we approximate:

$$\mathbb{E}\sum_{i=1}^n S(\boldsymbol{x}_i; \boldsymbol{\theta}) \approx \sum_{i=1}^n S(\boldsymbol{x}_i; \boldsymbol{\theta})|\{\boldsymbol{\theta} \in \Theta_{feasible}\} \geq nc$$

Therefore, with probability at least $1 - \delta$, we have

$$\sum_{i=1}^n S(\boldsymbol{x}_i; \boldsymbol{\theta}) \geq \mathbb{E}\sum_{i=1}^n S(\boldsymbol{x}_i; \boldsymbol{\theta}) - \sqrt{\frac{n\log(1/\delta)}{2}}$$

$$\geq nc - \sqrt{\frac{n\log(1/\delta)}{2}}$$

$$\implies \frac{1}{n}\sum_{i=1}^n S(\boldsymbol{x}_i; \boldsymbol{\theta}) \geq c - \sqrt{\frac{\log(1/\delta)}{2n}},$$

i.e. the model is improbable to collapse if the feasible region is non-negligible.

**Case 2 :** $g(\boldsymbol{\theta}^*) = a^k + \boldsymbol{b}^{k\top} \boldsymbol{S}(\boldsymbol{x}; \boldsymbol{\theta}) > 0$:

Similarly, we have

$$g(\boldsymbol{\theta}^*) \leq \beta \left( \left| \sum_{i=1}^n b_i^k \Delta_i \right| \right)^{-1} \frac{2\phi_{max} |\sum_{i=1}^n \Delta_i|}{\sum_{i=1}^n S(\boldsymbol{x}_i; \boldsymbol{\theta})}. \tag{15}$$

Let $Z = \sum_{i=1}^n b_i^k \Delta_i$, then $\mathbb{E}Z = \sum_{i=1}^n b_i^k \mathbb{E}\Delta_i = \mu_\Delta \sum_{i=1}^n b_i^k$.

By the triangle inequality, we have $\left| |Z| - |\mathbb{E}Z| \right| \leq |Z - \mathbb{E}Z|$. So, by the Hoeffding inequality, we obtain

$$P\left( \left| |Z| - |\mathbb{E}Z| \right| \geq t \right) \leq P(|Z - \mathbb{E}Z| \geq t)$$

$$\leq 2 \exp\left( -\frac{2t^2}{4L^2 \sum_{i=1}^n (b_i^k)^2} \right)$$

Let $t = L\sqrt{2 \sum_{i=1}^n (b_i^k)^2 \log \frac{2}{\delta}}$. Then with probability at least $1 - \delta$,

$$|Z| \geq |\mathbb{E}Z| + t = |\mu_\Delta \sum_{i=1}^n b_i^k| + t.$$

Substituting this into Eq. equation 15, we obtain

$$g(\boldsymbol{\theta}^*) \leq \beta \frac{2\phi_{max} |\sum_{i=1}^n \Delta_i|}{\left( |\mu_\Delta \sum_{i=1}^n b_i^k| + t \right) \sum_{i=1}^n S(\boldsymbol{x}_i; \boldsymbol{\theta})}$$

$$\leq \beta \frac{2\phi_{max} \sum_{i=1}^n |\Delta_i|}{\left( |\mu_\Delta \sum_{i=1}^n b_i^k| + t \right) \sum_{i=1}^n S(\boldsymbol{x}_i; \boldsymbol{\theta})}$$

$$\leq \beta \frac{2\phi_{max} L}{\left( |\mu_\Delta \sum_{i=1}^n b_i^k| + t \right) \frac{1}{n} \sum_{i=1}^n S(\boldsymbol{x}_i; \boldsymbol{\theta})}$$

As analyzed in **Case 1**, when the feasible region is non-negligible, $\frac{1}{n} \sum_{i=1}^n S(\boldsymbol{x}_i; \boldsymbol{\theta}) > c - \xi$ with high probability. Therefore,

$$g(\boldsymbol{\theta}^*) \leq \beta \frac{2\phi_{max} L}{\left( |\mu_\Delta \sum_{i=1}^n b_i^k| + t \right)(c - \xi)}$$

$$\text{Plug in } \beta \leq \frac{\xi(c - \xi)|\mu_\Delta|}{2\phi_{max} L},$$

$$\leq \frac{\xi |\mu_\Delta|}{|\mu_\Delta \sum_{i=1}^n b_i^k| + t}$$

$$\leq \frac{\xi |\mu_\Delta|}{|\mu_\Delta \sum_{i=1}^n b_i^k| + L\sqrt{2 \sum_{i=1}^n (b_i^k)^2 \log \frac{2}{\delta}}}$$

$$\leq \frac{\xi}{|\sum_{i=1}^n b_i^k| + \frac{L}{|\mu_\Delta|}\sqrt{2 \sum_{i=1}^n (b_i^k)^2 \log \frac{2}{\delta}}}$$

By Cauchy-Schawarz inequality, we have $\sqrt{\sum_{i=1}^n (b_i^k)^2} \geq \frac{|\sum_{i=1}^n b_i^k|}{\sqrt{n}}$, therefore,

$$g(\boldsymbol{\theta}^*) \leq \frac{\xi}{|\sum_{i=1}^n b_i^k| + \frac{L}{|\mu_\Delta|} \frac{|\sum_{i=1}^n b_i^k|}{\sqrt{n}}\sqrt{2 \log \frac{2}{\delta}}}$$

$$= \frac{\xi}{|\sum_{i=1}^n b_i^k| \left( 1 + \frac{L}{|\mu_\Delta|\sqrt{n}}\sqrt{\log \frac{2}{\delta}} \right)}$$

Put together **Case 1** and **Case 2**, we finish the proof.

$\square$

## C   IMPLEMENTATION DETAILS

### C.1   SELECTION OF $\hat{\phi}$

$\frac{\sum_{i=1}^n S(\boldsymbol{x}_i;\theta)\hat{\phi}(\boldsymbol{x}_i,a_i,y_i)}{\sum_{i=1}^n S(\boldsymbol{x}_i;\theta)}$ can be used to represent the ATE estimated by different methods. And the following $\hat{\phi}_{\text{iptw}}$ and $\hat{\phi}_{\text{aiptw}}$, correspond to the ATE estimated using IPTW and AIPTW, respectively. *inverse probability of treatment weighting* (IPTW) and *augmented inverse probability of treatment weighting* (AIPTW), respectively. To clarify, $\phi(\boldsymbol{x}_i,a_i,y_i)$ does not depend on the parameter $\boldsymbol{\theta}$, these estimates are separated from the parametric surrogate model $S$.

$$\hat{\phi}_{\text{iptw}}(\boldsymbol{x}_i,a_i,y_i) = \frac{a_i}{\hat{e}(\boldsymbol{x}_i)}y_i - \frac{1-a_i}{1-\hat{e}(\boldsymbol{x}_i)}y_i,$$

$$\hat{\phi}_{\text{aiptw}}(\boldsymbol{x}_i,a_i,y_i) = \hat{\mu}_1(\boldsymbol{x}_i) - \hat{\mu}_0(\boldsymbol{x}_i) + \frac{a_i}{\hat{e}(\boldsymbol{x}_i)}(y_i - \hat{\mu}_1(\boldsymbol{x}_i))$$

$$- \frac{1-a_i}{1-\hat{e}(\boldsymbol{x}_i)}(y_i - \hat{\mu}_0(\boldsymbol{x}_i)).$$

### C.2   SYNTHETIC DATA GENERATION

Let $\sigma_X, \sigma_Y, \rho \in \mathbb{R}$ be fixed constants, and let $\beta_1, \beta_\tau, \omega \in \mathbb{R}^p$. Draw $\boldsymbol{X}$ according to a multi-variate normal distribution, and $A, Y(0), Y(1)$ as follows:

$$\boldsymbol{X} \sim \mathcal{MVN}(0, \sigma_X^2[(1-\rho)I_p + \rho 1_p 1_p^T]), \tag{16}$$

$$A|\boldsymbol{X} \sim \text{Bernoulli}(\sigma(\boldsymbol{X}^T\omega)), \tag{17}$$

$$\epsilon \sim \mathcal{N}(0, \sigma_Y^2), \tag{18}$$

$$Y(0) = (\sin(10*\boldsymbol{X}) + 5*\boldsymbol{X}^2)^T\beta_1 + \epsilon, \tag{19}$$

$$Y(1) = (\sin(10*\boldsymbol{X}) + 5*\boldsymbol{X}^2)^T\beta_1 + \boldsymbol{X}^T\beta_\tau + \epsilon. \tag{20}$$

We set the parameters as follows:

$$p = 10, \quad \sigma_X = \sigma_Y = 0.1, \quad \rho = 0.3,$$
$$\beta_1 = [0, 0, 0, 0, 2, 0, 0, 0, 0, 0],$$
$$\beta_\tau = [0.5, 0.5, 0.5, 0.5, 0, 0, 0, 0, 0, 0],$$
$$\omega = [0, -1\cdot\tilde{\omega}, -1\cdot\tilde{\omega}, 1\cdot\tilde{\omega}, 1\cdot\tilde{\omega}, -2\cdot\tilde{\omega}, 0, 0, 0, 0].$$

The *imbalance parameter*, $\tilde{\omega} \geq 0$, scales the magnitude of the treatment assignment weight vector $\omega$. In particular, we generated two synthetic datasets with (1) no confounding bias ($\tilde{\omega} = 0$); and (2) high confounding bias ($\tilde{\omega} = 5$). As there is no limitation to generate synthetic data, we set up the total sample size for synthetic data as 5,000 to study the model performance under finite samples and use a balance 50/50 train-test split ratio. As for real-world datasets, we use a 70/30 train-test split.

### C.3   BASELINE IMPLEMENTATION

**CAPITAL** identifies a subgroup by maximizing its size while ensuring the CATE exceeds a pre-defined threshold. Since subgroup size is not directly controlled in this setting, we vary the CATE threshold and construct the group size vs. ATE curve for comparison.

**OWL** is originally designed for individual treatment rule estimation but can be viewed as a subgroup identification method without interpretability constraints. It assigns scores between 0 and 1, which we threshold to obtain subgroups of a desired size. We implement OWL using the DTRlearn2 R package.

| Dataset | Dragonnet | LR |
|---|---|---|
| Synthetic | 0.10 | 0.10 |
| MIMIC-IV | 2.32 | 1.57 |
| eICU | 1.08 | 1.10 |

Table 2: Number of unbalance features after reweighting by propensity scores obtained by propensity models

**VT** The original VT supports both binary and continuous outcomes, but the commonly used R package aVirtualTwins handles only binary outcomes. Since our experiments require both, we implemented our own VT following Foster et al. (2011): first estimating treatment effects with a random forest, and then fitting a regression tree to assign subgroup scores. Subgroups of different sizes are obtained by thresholding these scores.

**Dragonnet & CT & CF** We adopts the implementations from the Python package causalml.

### C.4 MODEL SELECTION FOR NUISANCE FUNCTION ESTIMATION

We consider LR and Dragonnet for estimating the propensity score model. We following the literature Zang et al. (2023) to use the number of unbalanced features after inverse propensity score weighing as the metric (the lower the better) to select the propensity score model. The results are shown in Table 2, which shows that LR performs better or equivalently than Dragonnet. Thus we select LR as the propensity score model for all datasets.

For the potential outcome model, we consider Dragonnet, CF, and CT, as they are the CATE estimators we are comparing against in subgroup identification. Theoretically, CF improves upon CT by reducing bias and producing smoother decision boundaries through aggregating CTs. As for Dragonnet and CF, prior work Kiriakidou & Diou (2022) has shown that Dragonnet outperforms CF.

While many other methods exist for nuisance function estimation, our goal is to demonstrate MOSIC's flexibility rather than prescribe a specific estimator. If more effective models are developed or if a particular method performs better in a given setting, we encourage adapting those for nuisance function estimation.

### C.5 HYPERPARAMETER TUNING

CAPITAL optimizes the policy tree depth with $max\_depth \in \{2, 3\}$. VT selects decision tree depth for subgroup identification from $max\_depth \in \{3, 5, 7, 10\}$. Similarly, CT uses $max\_depth \in \{3, 5, 7, 10\}$, while CF considers both tree depth $max\_depth \in \{3, 5, 7, 10\}$ and the number of trees $num\_tree \in \{5, 10, 20, 50, 100\}$. Moreover, Dragonnet tunes the hidden layer size with $hidden\_size \in \{50, 100, 200\}$. Finally, the hyperparameters for our method include $\beta \in \{10^{-2}, 10^{-3}, 10^{-4}, 10^{-5}\}$ and those depending on the implementation of the subgroup identification model. For MOSIC-MLP, we tune the hidden layer size with $hidden\_size \in \{50, 100, 200\}$.

## D SIGNIFICANCE TESTS ON READ-WORLD DATA

Tables D.0.3 and D.0.4 present the p-values comparing MOSIC ($\alpha = 0.01$) against baseline methods, corresponding to the hypothesis test that MOSIC is different than the baselines. These results directly support the performance trends shown in Figure 2. For instance, in eICU experiments with c=0.5:

- MOSIC achieves significantly better ATE than Dragonnet (p=0.0057)

- MOSIC maintains significantly fewer unbalanced features than Dragonnet (p=9.2E-04)

|  |  | $c = 0.4$ | $c = 0.5$ | $c = 0.6$ | $c = 0.7$ | $c = 0.8$ |
|---|---|---|---|---|---|---|
| ATE | CT | 0.71 | 0.065 | 0.0011 | 0.0046 | 0.025 |
|  | CF | 0.19 | 0.0048 | 0.00019 | 0.0012 | 0.0057 |
|  | CA* | - | - | - | 3.5E-05 | - |
|  | DR* | 0.30 | 0.0057 | 5.5E-04 | 0.011 | 0.11 |
|  | OWL | 0.27 | 0.0019 | 0.0010 | 0.084 | 0.16 |
|  | VT | 0.38 | 0.07 | 0.039 | 0.31 | 0.46 |
| Balance | CT | 0.16 | 0.58 | 0.56 | 0.65 | 0.04 |
|  | CF | 0.35 | 0.12 | 0.48 | 0.47 | 0.16 |
|  | CA* | - | - | - | 0.59 | - |
|  | DR* | 0.022 | 9.2E-04 | 0.14 | 0.33 | 0.066 |
|  | OWL | 0.0088 | 0.0077 | 0.13 | 0.22 | 0.046 |
|  | VT | 0.0011 | 0.0060 | 0.036 | 0.23 | 0.39 |

Table D.0.3: eICU: p-values for ATE and feature balance comparisons between MOSIC and baseline methods. CA*: CAPITAL; DR*: Dragonnet.

|  |  | $c = 0.5$ | $c = 0.6$ | $c = 0.7$ | $c = 0.8$ |
|---|---|---|---|---|---|
| ATE | CT | - | - | - | 0.02 |
|  | CF | 0.42 | 0.68 | 0.35 | 0.16 |
|  | CA* | 0.021 | - | 0.30 | 0.72 |
|  | DR* | 0.039 | 0.10 | 0.21 | 0.30 |
|  | VT | 0.10 | 0.16 | 0.15 | 0.15 |
| Balance | CT | - | - | - | 0.63 |
|  | CF | 0.19 | 5.1E-04 | 1.3E-04 | 1.1E-04 |
|  | CA* | 6.3E-07 | - | 0.17 | 3.3E-05 |
|  | DR* | 0.062 | 0.80 | 0.28 | 0.48 |
|  | VT | 0.033 | 0.0020 | 0.018 | 0.20 |

Table D.0.4: MIMIC: p-values for ATE and feature balance comparisons between MOSIC and baseline methods. CA*: CAPITAL; DR*: Dragonnet.

# E  ADDITIONAL EXPERIMENT RESULTS

## E.1  UNCERTAINTY QUANTIFICATION

To quantify uncertainty, we computed the asymptotic 95% confidence intervals based on the closed-form influence function of the AIPTW estimator. Figure E.1.1 reports the distribution of CI widths across 100 independent train–test splits, showing the anticipated increase in uncertainty as subgroup size decreases.

Additionally, we report the Risk Ratio (RR) and E-values to measure the robustness of our subgroup results with respect to unmeasured confounders.

## E.2  ADDITIONAL BASELINES

We further compare against DR-learner, R-learner, BART, and an overlap-weighted variant of MOSIC (MOSIC-OW). We adopted the Python package causalml to implement DR-learner and R-learner, using Random Forest as their base learner. For BART, we adopted the R package bart-Cause. For hyperparameters, DR-learner and R-learner consider $max\_depth \in \{3, 5, 7, 10\}$, and BART considers the number of prior standard deviations $k \in \{1, 2, 3\}$. The hyperparameter tuning procedures are the same as described in Section 5.1.

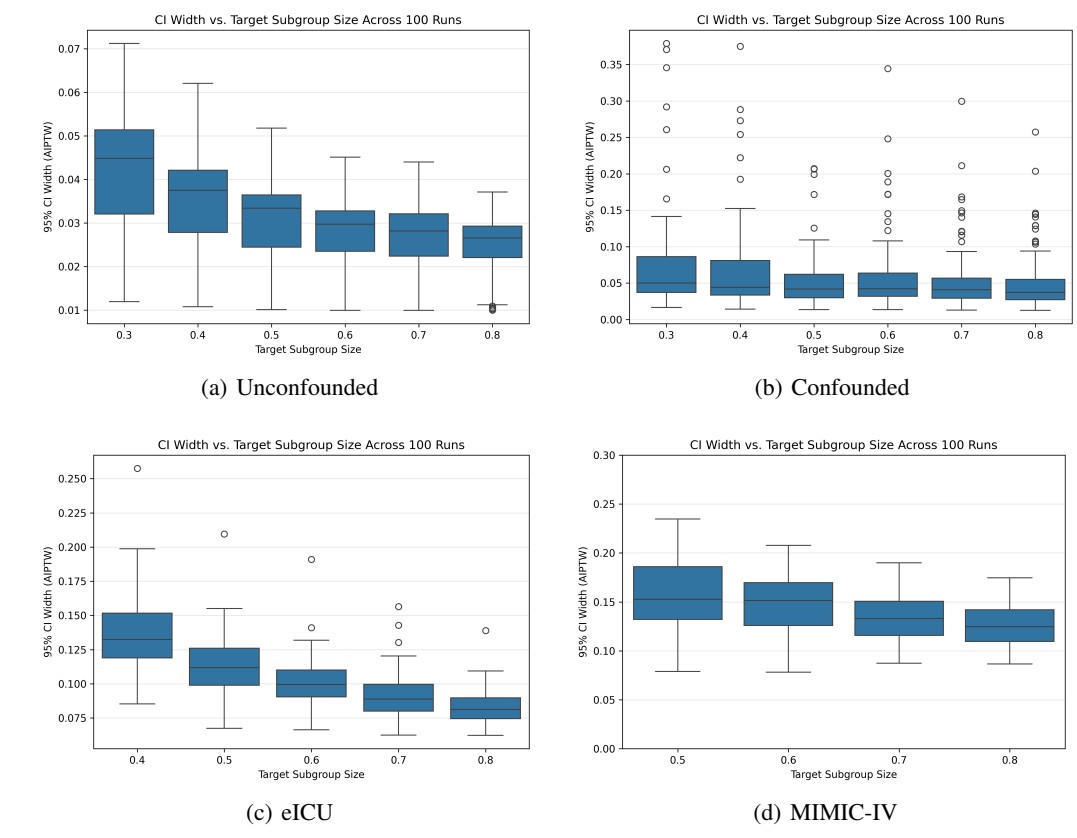

(a) Unconfounded

(b) Confounded

(c) eICU

(d) MIMIC-IV

Figure E.1.1: CI width.

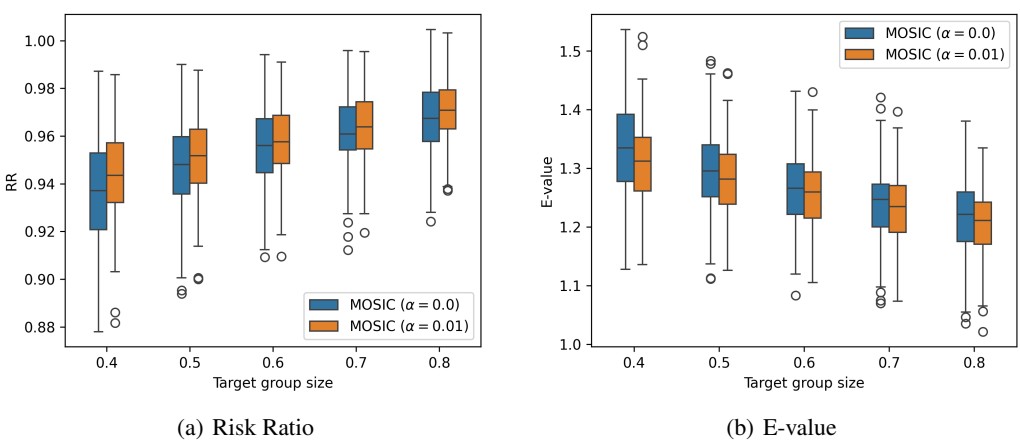

(a) Risk Ratio

(b) E-value

Figure E.1.2: Sensitivity analysis on unmeasured confounding

## E.3 INCORPORATING SAMPLE-SPLITTING

To assess whether internal sample-splitting improves subgroup learning, we implemented 5-fold cross-fitting on the training data: nuisances were trained on 4 folds and used to generate CATEs on the held-out fold, and the subgroup model was trained on these cross-fitted CATEs. Evaluation remained on the untouched test set, where nuisances were refit on the full training set before computing AIPTW. The results (Figure E.3.1) are similar to our original pipeline, suggesting that the benefit of decoupling nuisance and subgroup estimation is limited in our setting, likely because further splitting of a modest dataset weakens nuisance estimation.

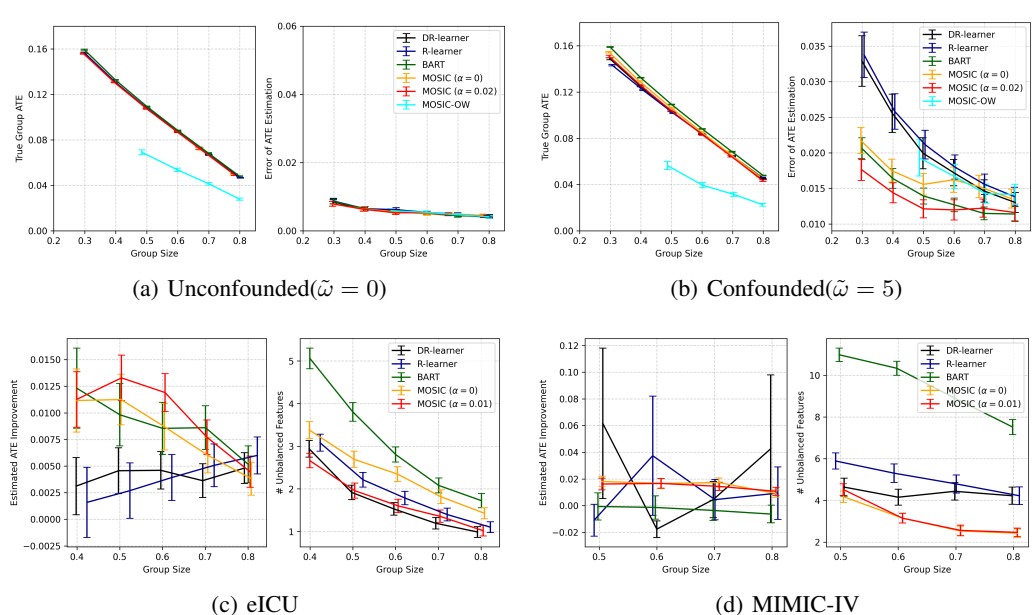

(a) Unconfounded($\tilde{\omega} = 0$)  (b) Confounded($\tilde{\omega} = 5$)

(c) eICU  (d) MIMIC-IV

Figure E.2.1: Estimated ATE and the number of unbalanced features on real-world datasets. MOSIC-OW performs substantially worse in synthetic experiments, so we omit it from real-world comparisons.

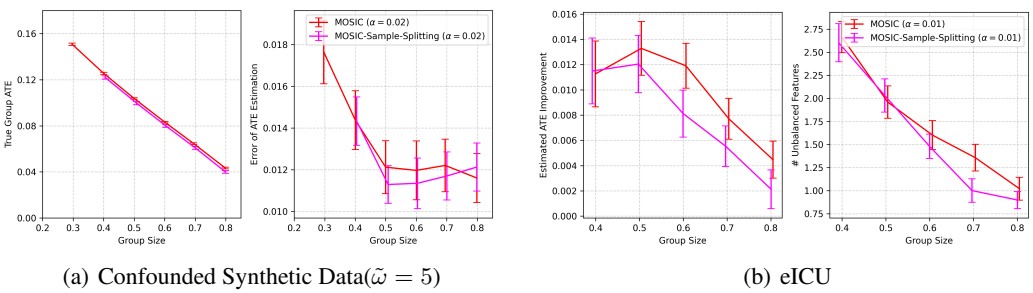

(a) Confounded Synthetic Data($\tilde{\omega} = 5$)  (b) eICU

Figure E.3.1: Comparison of MOSIC with cross-fitted CATE estimates.

## E.4 OVERLAP CONSTRAINT ON SYNTHETIC DATA

We numerically verify that MOSIC can indeed accommodate the overlap constraint (Figure E.4.1). To assess its effect on estimation error, we conduct experiments on the confounded synthetic data with varying $\alpha$ values. Figure E.4.1(a) is generated by a random instance of the 100 random train-test splits presented in Figure E.4.1(b). In Figure E.4.1(a), each dot represents an individual, with the x-axis denoting the estimated propensity score, the y-axis representing the true individual treatment effect (ITE), and the vertical lines indicating the desired overlap threshold. Without overlap constraints, MOSIC selects patients with the highest ITEs.

With overlap constraints (Figure E.4.1(a), left), MOSIC continues to select patients with large ITEs but systematically excludes those who violate the overlap constraint ($\hat{e}(x) < 0.05$ or $\hat{e}(x) > 0.95$). Figure E.4.1(b) further quantifies the effect of excluding patients with limited overlap by reporting the estimation error and the number of unbalanced features.

As the desired overlap threshold $\alpha$ increases, the estimation error of the subgroup ATE decreases, suggesting that enforcing stronger overlap improves estimation reliability. Meanwhile, stronger overlap also correlates with a lower number of unbalanced features.

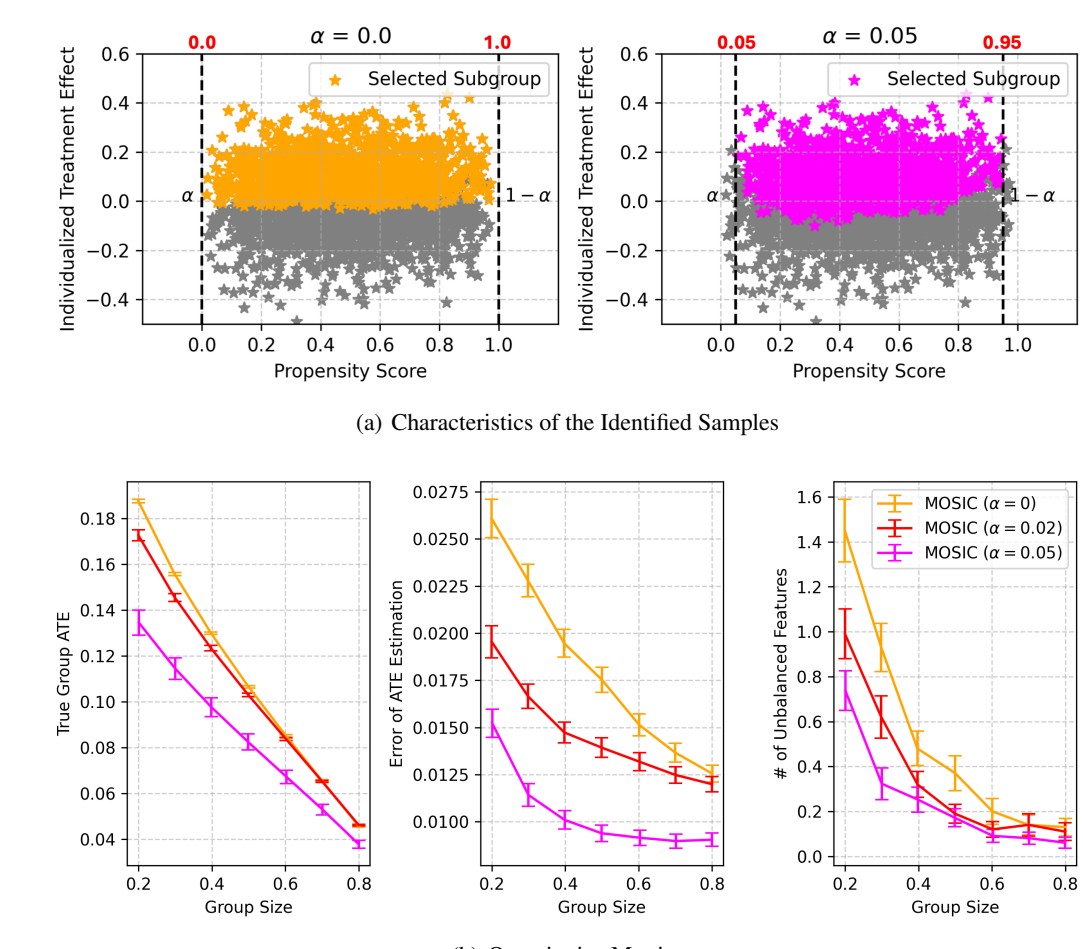

(a) Characteristics of the Identified Samples

(b) Quantitative Metrics

Figure E.4.1: Results on synthetic data with confounding bias ($\tilde{\omega} = 5$), obtained using MOSIC with varying $\alpha$.

### E.5 ADDITIONAL ANALYSIS OF OVERLAP ON REAL-WORLD DATA

We evaluated overlap on the held-out test sets and report the proportions of samples with estimated propensity scores falling outside [0.01,0.99], [0.02,0.98], and [0.05,0.95] within the subgroups selected on the test sets (Figure E.5.1). When no overlap constraint is imposed ($\alpha = 0$), the proportion of low-overlap samples in the selected subgroup closely matches that of the full test set. In contrast, when overlap constraints are activated ($\alpha > 0$), these proportions decrease substantially across all thresholds, demonstrating that MOSIC's overlap constraint effectively improves overlap in the selected subgroups during evaluation as well as training.

### E.6 ADDITIONAL ANALYSIS OF FEATURE IMBALANCE ON REAL-WORLD DATA

SMD>0.1 is a stricter threshold to evaluate feature imbalance and commonly used in epidemiology studies. However, Austin (2009) notes that "For modest sample sizes, one could expect standardized differences that exceed 0.20 (20 percent) even when the propensity-score model was correctly specified." Given that both of our real-world cohorts are relatively small (13,361 patients in eICU and 6,516 in MIMIC), we chose SMD > 0.2 as the primary threshold to reduce the risk of flagging spurious imbalance driven by sample size limitations. This threshold is also frequently used in epidemiology studies.

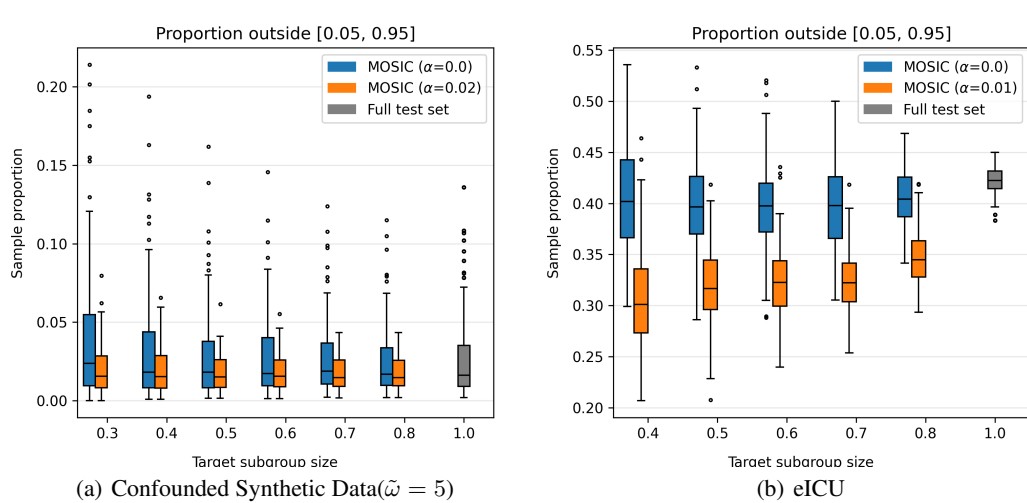

(a) Confounded Synthetic Data($\tilde{\omega} = 5$)  (b) eICU

Figure E.5.1: Overlap Evaluation on Test Set.

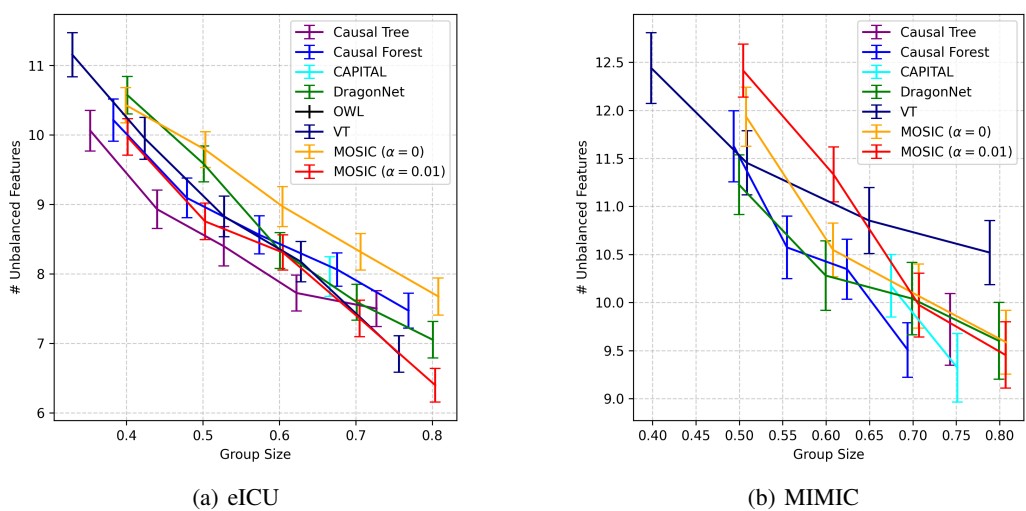

(a) eICU  (b) MIMIC

Figure E.6.1: Number of unbalanced features using SMD>0.1 as threshold.

We now report the results using the stricter SMD > 0.1 criterion, and it does not alter our main conclusions. Although the absolute number of unbalanced covariates increases across all methods due to inherent data size limitations, the relative comparison remains the same: MOSIC achieves comparable balance while yielding substantially higher subgroup ATEs (Figure E.6.1 and Figure 2). (While DragonNet demonstrates significantly lower feature imbalance, our method obtain significantly higher subgroup ATE improvement than DragonNet, as shown in Figure 2b.)

### E.7 EXTENSION TO SAFETY, BUDGET, AND FAIRNESS CONSTRAINT ON SYNTHETIC DATA

To demonstrate MOSIC's flexibility to handle additional constraints, we extend the synthetic data generation process described in Appendix C.2 by introducing a safety, budget, and fairness constraint. In particular:

- Safety constraint: Following Doubleday et al. (2022), each sample is assigned a risk score $r_i = 1/(1 + \exp(10 * x_i[10] + 1))$. We require the average risk of the selected subgroup to

|      | c=0.4  | c=0.5  | c=0.6 | c=0.7 | c=0.8  |
|------|--------|--------|-------|-------|--------|
| CA*  | –      | –      | 0.36  | –     | –      |
| CF   | 0.54   | 0.39   | 0.50  | 0.050 | 0.0024 |
| DT*  | 0.82   | 0.65   | 0.81  | 0.32  | 0.0022 |
| DR   | 0.11   | 0.027  | 0.93  | 0.52  | 0.069  |
| VT   | 0.0044 | 0.0033 | 0.18  | 0.04  | 0.21   |

Table E.6.1: eICU: p-values for feature balance comparisons between MOSIC and baseline methods using SMD>0.1 as threshold.CA*: CAPITAL; DR*: Dragonnet.

|      | c=0.5   | c=0.6 | c=0.7 | c=0.8 |
|------|---------|-------|-------|-------|
| CA*  | 1.8e-10 | –     | 0.67  | –     |
| CF   | 0.091   | 0.081 | 0.41  | 0.91  |
| CT   | –       | –     | –     | 0.60  |
| DR*  | 0.0049  | 0.024 | 0.89  | 0.78  |
| VT   | 0.095   | 0.79  | 0.067 | 0.028 |

Table E.6.2: MIMIC: p-values for feature balance comparisons between MOSIC and baseline methods using SMD>0.1 as threshold.CA*: CAPITAL; DR*: Dragonnet.

be no greater than 0.05:

$$\frac{\sum_{i=1}^{n} \mathbb{1}(S(x_i) > 0.5)r_i}{\sum_{i=1}^{n} \mathbb{1}(S(x_i) > 0.5)} \le 0.05.$$

- Budget constraint: Following Qiu et al. (2022), each sample is assigned a treatmentd cost value $cost_i = (x_i[3] + 5)/5$. The total cost of the selected subgroup must not exceed half the cost of treating the entire population (assuming a unit cost per sample):

$$\sum_{i=1}^{n} \mathbb{1}(S(x_i) > 0.5)cost_i \le 0.5 * n.$$

The test contains 2500 samples, so the total cost limit $0.5 * n = 1250$ in this case.

- Fairness constraint: Let a binary sensitive attribute: $sens_i = \mathbb{1}(x[3] > 0.5)$. We adopt the conditional statistical parity metric Mehrabi et al. (2021). In our setting, this corresponds to maintaining a sensitive-group proportion of 0.5 in the selected subgroup.:

$$\left| \frac{\sum_{i=1}^{n} \mathbb{1}(S(x_i) > 0.5)sens_i}{\sum_{i=1}^{n} \mathbb{1}(S(x_i) > 0.5)} - 0.5 \right| \le 0.01,$$

where we allow a violation of 0.01. We can then convert this to two ratio-form constraints:

$$-0.01 \le \frac{\sum_{i=1}^{n} \mathbb{1}(S(x_i) > 0.5)sens_i}{\sum_{i=1}^{n} \mathbb{1}(S(x_i) > 0.5)} - 0.5 \le 0.01.$$

In addition to the group size constraint ($c = 0.5$) and overlap constraint($\alpha = 0.02$), we progressively add the following constraints: 1) Plus safety constraint; 2) Plus safety and budget constraint; 3) Plus safety, budget, and fairness constraint. Results in Table E.7.1 show that MOSIC can effectively enforce all of these constraints.

### E.8 EXTENSION TO SAFETY CONSTRAINT ON eICU

The Glasgow Coma Scale (GCS), a core component of the Sequential Organ Failure Assessment (SOFA) score, measures level of consciousness, with lower scores indicating more severe dysfunction. We targeted patients with GCS < 6 because this threshold represents the most severe central nervous system dysfunction in the SOFA score, the most commonly-used ICU metric. Without a constraint on the GCS score, we observed that the selected subgroup contains a nontrivial number

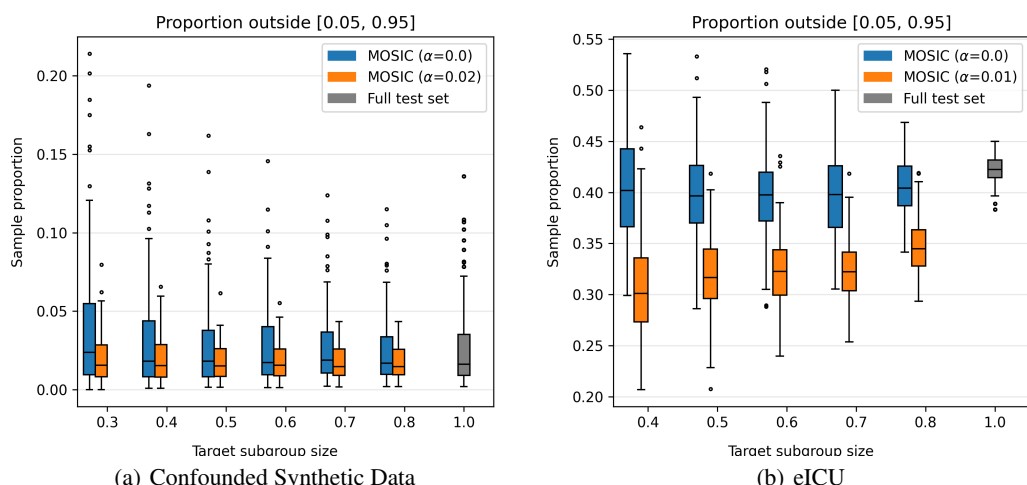

(a) Confounded Synthetic Data        (b) eICU

Figure E.6.2: Overlap Evaluation on Test Set

Table E.7.1: Performance on synthetic data ($\tilde{\omega} = 5$) under multiple additional constraints. Constraints: 1) Safety: Average Risk $\leq 0.05$; 2) Budget: Total Cost $\leq 1250$; 3) Fairness: |Sensitive Group Ratio - 0.5| $\leq 0.01$.

| Metric | Group Size & Overlap | Group Size & Overlap & Safety | Group Size & Overlap & Safety & Budget | Group Size & Overlap & Safety & Budget & Fairness |
|---|---|---|---|---|
| Group Size | 0.50 ± 0.01 | 0.50 ± 0.05 | 0.48 ± 0.05 | 0.49 ± 0.02 |
| # Unbalance | 0.14 ± 0.40 | 0.20 ± 0.45 | 0.23 ± 0.57 | 0.23 ± 0.45 |
| True CATE | 0.10 ± 0.01 | 0.10 ± 0.01 | 0.09 ± 0.01 | 0.09 ± 0.01 |
| ATE Error | 0.01 ± 0.01 | 0.01 ± 0.01 | 0.01 ± 0.01 | 0.01 ± 0.01 |
| Average Risk | 0.08 ± 0.01 | **0.05 ± 0.01** | **0.05 ± 0.01** | **0.05 ± 0.01** |
| Total Cost | 1377.41 ± 41.57 | 1365.72 ± 144.95 | **1252.78 ± 132.59** | 1268.40 ± 42.44 |
| Sensitive Group Ratio | 0.45 ± 0.03 | 0.45 ± 0.03 | 0.46 ± 0.03 | **0.49 ± 0.02** |

of patients with GCS< 6. As observational studies have shown that glucocorticoids may exacerbate neural system damage, such a subgroup raises a safety concern. Therefore, we required that the proportion of patients with GCS < 6 remain below 0.05. We did not enforce a strict exclusion (i.e., a threshold of 0) because it is possible that some patients with severe neural dysfunction could still derive meaningful benefit from the treatment.

To impose the interpretability requirement, we implement MOSIC-DT for this setting. Similarly, we run experiments on 100 random train-test splits and use 5-fold cross-validation to select the tree depth among {3,5,7}. Results in Table 1 show MOSIC can effectively exclude patients with GCS< 6. Additionally, an example (Figure 5) shows that MOSIC indeed learned to exclude such high-risk patients.

We additionally run MOSIC-MLP with and without the GCS< 6 constraints and run post-hoc SHAP-value analysis to evaluate the feature contributions. The dominant features identified by SHAP are generally consistent with the rule structure revealed by the decision tree. In particular, the SHAP values of the GCS features was ambiguous before we enforced the constraint that avoids patients with GCS < 6, but it became clearly separated after we enforced it, aligning well with our intention.

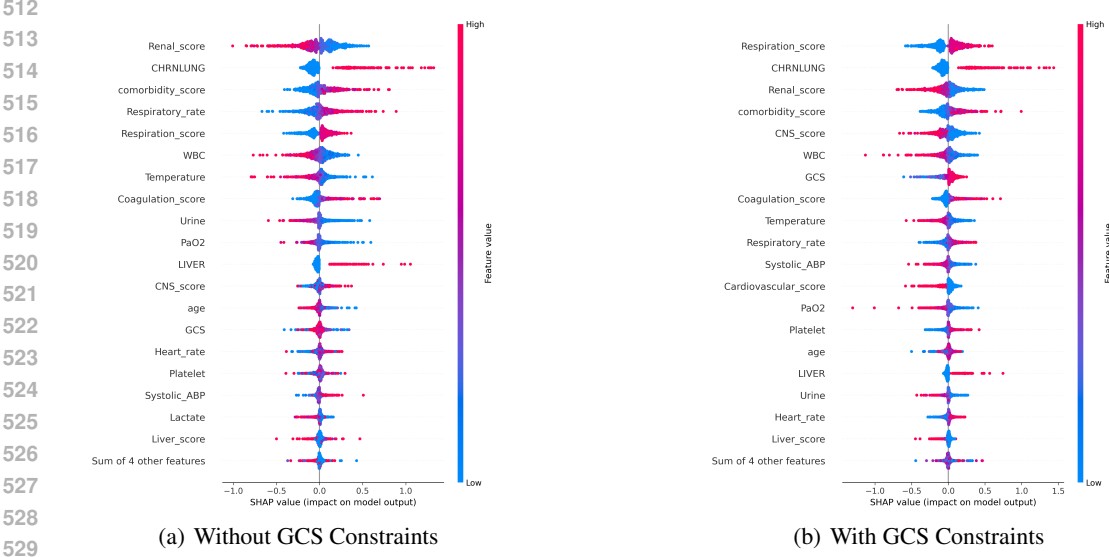

(a) Without GCS Constraints  (b) With GCS Constraints

Figure E.8.1: SHAP values of MOSIC-MLP

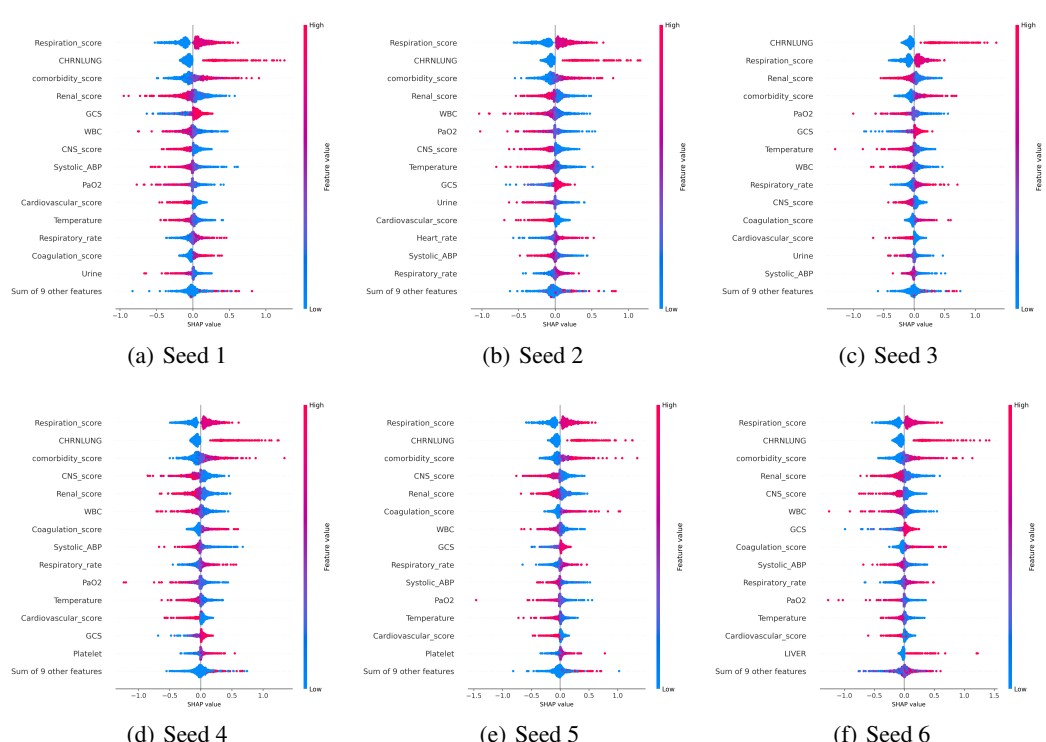

(a) Seed 1  (b) Seed 2  (c) Seed 3

(d) Seed 4  (e) Seed 5  (f) Seed 6

Figure E.8.2: Model Stability across different initialization

We further evaluate subgroup stability across different initializations. For the same train–test split, we reran MOSIC-MLP six times with the same constraints but different random initializations and examined whether the resulting feature contributions remained consistent. As shown in Figures E.8.2, the SHAP value patterns are highly similar across runs, suggesting that the learned subgroups are stable with respect to initialization and all align well with the safety constraint.

### E.9 TRAINING DYNAMICS

Figure E.9.1 shows the training loss and constraint violation during training. In the right panel, the plotted values represent the total magnitude of constraint violation: positive values indicate that one or more constraints are violated, while a value of 0 indicates that all constraints are satisfied.

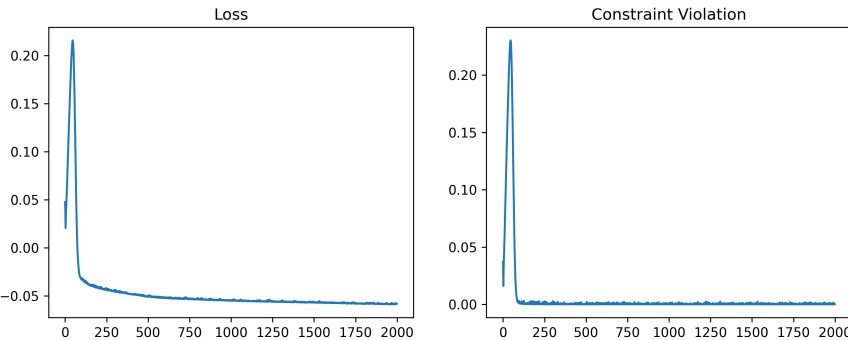

Figure E.9.1: Example training dynamics for a single run on the eICU dataset.

## F EXAMPLE OF GDA INSTABILITY

Consider a scenario with a single group size constraint $(c - \frac{1}{n} \sum_{i=1}^{n} S(\boldsymbol{x}_i; \boldsymbol{\theta}))$ and its Lagrange multiplier $\lambda_{n+1}$. Initially, the constraint is violated because the group size is below the threshold, leading to a positive gradient that increments $\lambda_{n+1}$ at each iteration. Once the constraint is met, the gradient becomes negative, driving $\lambda_{n+1}$ toward 0. However, due to the small learning rate, $\lambda_{n+1}$ remains nonzero in subsequent iterations, continuing to penalize the objective and lead to artificially shrunk feasible regions. This instability extends to multiple constraints, causing the algorithm to oscillate between a strictly feasible but suboptimal $\boldsymbol{\theta}$ and an optimal but infeasible $\boldsymbol{\theta}$, undermining convergence.

## G EVALUATION ON BINARY SUBGROUP SETTING

Unlike approaches that assume the existence of two or a finite number of subgroups with distinct ATEs, our study focuses on identifying a subset of the population with maximal ATE under real-world constraints, without making structural assumptions about the underlying heterogeneity. This design enables our method to generalize to continuous or complex heterogeneity. Nevertheless, we recognize that such structural assumptions, such as that binary subgroups with distinct ATE exist, are plausible in certain real-world scenarios. In such cases, the real-world constraint will naturally introduce a trade-off for identification performance.

To illustrate, we modify the DGP of synthetic data by replacing $Y(1)$ in Appendix C.2 to:

$$Y(1) = (\sin(10 * \boldsymbol{X}) + 5 * \boldsymbol{X}^2)^T \beta_1 + \mathbb{1}(\boldsymbol{X}^T > 0.05)\beta_\tau + \epsilon.$$

That says, only patients with covariates $> 0.05$ at positive $\beta_\tau$ indices receive a positive effect; others receive none. This yields a positive subgroup comprising 68% of samples. Using this DGP, we test MOSIC with beta = 1e-5, alpha = 0, and c in $\{0.6, 0.7, 0.8\}$. The precision and recall of subgroup identification are evaluated. Results in Table G.0.1 are reported as mean $\pm$ standard deviation over 100 runs. Performance aligns with theory: precision/recall degrade when $c$ (the group size constraint) exceeds/falls below the true subgroup size, reflecting constraint-driven trade-offs.

| c | ATE | Group Size | Precision | Recall |
|---|-----|-----------|-----------|--------|
| 0.6 | 0.92±0.07 | 0.60±0.01 | 0.93±0.05 | 0.83±0.05 |
| 0.7 | 0.84±0.05 | 0.70±0.01 | 0.88±0.04 | 0.92±0.04 |
| 0.8 | 0.75±0.04 | 0.80±0.01 | 0.80±0.03 | 0.96±0.04 |

Table G.0.1: Performance of binary subgroup identification

| c | ATE | Group Size | Type I error |
|---|-----|-----------|--------------|
| 0.4 | $-0.0011 \pm 0.0401$ | $0.3991 \pm 0.0141$ | 0.12 |
| 0.6 | $-0.0018 \pm 0.0298$ | $0.6020 \pm 0.0141$ | 0.00 |
| 0.8 | $-0.0000 \pm 0.0212$ | $0.8038 \pm 0.0121$ | 0.00 |

Table H.0.1: Type I error under different constraints for group size.

## H  EVALUATION OF TYPE I ERROR

Although our primary focus is on multi-constraint subgroup identification rather than statistical inference, we also evaluate Type I error of the identified subgroup. We use a data-splitting approach to test whether identified subgroups arise spuriously under the null hypothesis, where all individuals have zero treatment effect. Using data splitting, Type I error of the selected subgroup can be evaluated after we build the subgroup assignment model. Specifically:

1. Split the dataset (e.g., 50-50) into training (subgroup selection) and holdout (inference);

2. Train MOSIC on the training set to learn the subgroup model;

3. Apply the model to the holdout set to identify the subgroup, then compute its subgroup ATE, denoted as $ATE_{hold\_out}$

4. Test the null hypothesis on the holdout set:

   (a) Construct the distribution of subgroup ATE under the null (here by directly sampling from the test distribution, with bootstrap subsample size equaling the target subgroup size specified by the parameter $c$)

   (b) Compare $ATE_{hold\_out}$ to this distribution, and determine whether the null hypothesis should be rejected.

5. Repeat steps 1-4 on additional synthetic data instances, aggregate results, and estimate type-I error.

We generated 100 synthetic datasets using the same DGP in Appendix C.2 except that $\beta_\tau$ is set to **0**. This aligns with the null hypothesis: all individuals have zero treatment effect. For each instance, we set the bootstrap iterations to 10000. Type I error rate is computed as the proportion of instances in which the null hypothesis is rejected by 5% significance level.

As shown in Table H.0.1, the Type I error rate increases when the parameter $c$ is small, consistent with theoretical expectations. Smaller subgroups lead to higher variance in ATE estimates, highlighting a fundamental trade-off between real-world constraints and statistical reliability.

## I  RUNTIME ANALYSIS

Each method is run three times, and average run time is reported; these experiments are run on CPU only to reflect clinical computing environments with limited resources. Since MOSIC's nuisance estimation step adopted DragonNet and Logistic regression, it shares the same nuisance-fitting cost as DragonNet (Logistic Regression is negligible). Hence, the reported runtime for MOSIC reflects only the additional cost of the optimization step. As shown in Table I.0.1, this overhead is small relative to nuisance estimation, indicating that computation is unlikely to be a deployment bottleneck.

| Method | N = 1000 | N = 3000 | N = 10000 | N = 30000 |
|---|---|---|---|---|
| DragonNet | 15.77 | 25.86 | 58.22 | 151.32 |
| CT | 0.00 | 0.01 | 0.03 | 0.06 |
| CF | 0.02 | 0.05 | 0.16 | 0.41 |
| VT | 0.07 | 0.16 | 0.37 | 0.67 |
| OWL | 0.20 | 0.20 | 0.73 | 1.83 |
| CAPITAL | 1.37 | 7.29 | 25.33 | 89.57 |
| MOSIC (Constraint: Size + Overlap) | 1.64 | 1.97 | 1.67 | 2.24 |
| MOSIC (Constraint: Size + Overlap + Safety) | 1.50 | 1.78 | 1.90 | 2.53 |
| MOSIC (Constraint: Size + Overlap + Safety + Budget) | 2.34 | 1.78 | 1.74 | 2.43 |
| MOSIC (Constraint: Size + Overlap + Safety + Budget + Fairness) | 2.35 | 2.04 | 1.64 | 2.44 |

Table I.0.1: Runtime (seconds) of each method across different sample sizes. MOSIC shares the same first-stage nuisance fitting as DragonNet; reported times reflect MOSIC's second-stage optimization only.

