# OpenReview forum: "MOSIC: Model-Agnostic Optimal Subgroup Identification with Multi-Constraint for Improved Reliability"
_ICLR.cc/2026/Conference — Submitted to ICLR 2026_

### Official Review · Reviewer_puYb · 2025-10-27

**Soundness:** 3
**Presentation:** 2
**Contribution:** 3
**Rating:** 6
**Confidence:** 3

**Summary:**

This paper proposes MOSIC, a model-agnostic optimization framework for treatment subgroup identification under multiple real world constraints, including minimum subgroup size, propensity score overlap, and optionally safety or fairness constraints. Unlike standard two-stage approaches that first estimate CATE and then apply post-hoc thresholding or filtering, MOSIC directly formulates subgroup identification as a constrained optimization problem. Experiments on both synthetic datasets and real data show that MOSIC consistently achieves higher subgroup treatment effects while satisfying the specified constraints.

**Strengths:**

**Good Motivation**

The paper transfers the subgroup identification problem from the standard two-stage CATE estimation and filtering into an optimization problem with constraints. The idea is elegant, and this kind of shift is decision-focused and real-world aware, aligning with the emerging trend in causal machine learning.

**Theoretical Proof**

I like that the paper provides several theoretical lemmas to support their algorithms, e.g., the strict local min–max point in Lemma 2. I didn't check the full details of the proof, but they seem correct (although I could be mistaken). The authors also provide experiments on both synthetic and real-world data to show the proposed method’s performance compared with baselines.

**Weaknesses:**

My main concern is about the practical utility of the proposed methods. The reason why the standard two-stage CATE methods are widely accepted is because they are easy to understand, easy to use, and easy to interpret. While I appreciate the theoretical and empirical results that MOSIC presents in the paper, the key question is: is the gain worth the additional complexity?

**Easy to understand?**

The idea is straightforward: shifting from estimation to optimization. However, the transition from Problem I to Problem IV sometimes feels reactive rather than intentional. Section 4 in particular reads like a narrated debugging process (“we tried X but it was unstable, so we added Y...”), rather than a clean forward design narrative. This hurts clarity, especially for theoretical readers.

**Easy to use?**

There is no discussion of computational cost or efficiency. The two real-world datasets used have sample sizes of 13,361 and 6,516, but there is no reporting of runtime or resource overhead relative to simpler two-stage baselines. This omission limits the reader’s ability to judge whether the method is practical for real-time or large-scale deployment. And for many clinicians who may need to use this method, will it still be practical if they do not have sufficient computational resources?

**Easy to interpret?**

While the method supports decision-tree backbones in Section 5.3, the paper never shows a concrete learned subgroup rule. For example, a clinician might prefer an interpretable patient subgroup like: “patients aged > 60 should not take the drug.” But this kind of explicit rule is missing from the paper. For deployment and clinical trust, this is a significant gap.

**Questions:**

1. Line 156: The paper claims that the method naturally accommodates more general linear and ratio-form constraints. Could the authors clarify what kinds of constraints can be naturally supported by MOSIC, and what kinds cannot? Additionally, how does the computational or optimization complexity scale as more constraints are added, does it grow linearly, or is there a risk of exponential blow-up? I also wonder would there be a finite-sample guarantees with constraints?

2. I understand that strict local min–max is a standard target for nonconvex optimization. However, since we are dealing with patient data, robustness is a critical concern. You mention running experiments over 100 random train/test splits, but do you also evaluate stability across different random initialization seeds of the optimizer (not just different data splits)? In other words, do different seeds lead to meaningfully different subgroups being selected, even if the average ATE is similar? Is the identified subgroup S(x) stable across initializations?

3. As mentioned in the weaknesses section, providing runtime and interpretability analysis would significantly strengthen the paper, particularly for clinical decision. Even an approximate runtime comparison against two-stage baselines, and at least one example of a human-readable decision-tree subgroup rule, would be very helpful.

---

> ### Author Response · Authors · 2025-11-23
>
> We thank the reviewer for their thoughtful comments; our responses are provided below.
>
> **W1**
> We have revised Section 4.2 to present a clearer, forward-designed formulation and removed the debugging-style narrative to improve readability for theoretical audiences.
>
> **W2 \& Q1 \& Q3**
> We thank the reviewer for raising this important point. For runtime with respect to the number of constraints, each constraint contributes one evaluation per update. The constraint families considered in our work (linear and ratio-form constraints) require at most $n$ multiplications, where $n$ is the number of samples. Thus, the per-iteration cost of computing the loss is $O(nm)$, where $m$ is the number of constraints. In our problem formulation, there are n overlap constraints, but each overlap constraint contributes only one multiplication, so the overall computation cost for overlap constraints is $O(n*1)= O(n)$ rather than $O(n^2)$. As a result, even though the number of constraints is large, the per-iteration cost remains small.
>
> Additionally, we have now included an empirical runtime analysis for MOSIC and all baselines in Appendix I (line 1711): Each method is run three times, and average run time is reported; these experiments are run on CPU only to reflect clinical computing environments with limited resources.
>
> Since MOSIC's nuisance estimation step adopted DragonNet and Logistic regression, it shares the same nuisance-fitting cost as DragonNet (Logistic Regression is negligible). Hence, the reported runtime for MOSIC reflects only the additional cost of the optimization step. As shown in Table I.0.1, this overhead is small relative to nuisance estimation, indicating that computation is unlikely to be a deployment bottleneck.
>
> For context, two-stage baselines such as DragonNet, CT, and CF produce only CATE rankings and do not provide explicit subgroup decision rules, limiting their direct usability in clinical workflows. CAPITAL is the closest comparable method, but it supports only a single constraint and relies on a combinatorial tree search with worst-case complexity $O(n^k)$, where k is the tree depth. Thus,  building a trees deeper than 2 or 3 is impractical for CAPITAL.
>
> Overall, our results show that MOSIC introduces minimal computational burden beyond standard nuisance estimation, while providing a flexible and constraint-aware subgroup model. This supports the practicality of MOSIC in real-world clinical settings, even with limited computing resources.
>
> **W3 \& Q3**
> We thank the reviewer for this helpful suggestion. We clarify that a human-readable decision-tree subgroup rule was already included in our original submission (previously in the appendix). Following the reviewer’s feedback, we have moved this example into the main text for better visibility (Figure 5, line 514).
>
>
> **Q1**
> MOSIC’s support for linear and ratio-form constraints is discussed in line 305-311. Linear examples include the overlap constraint emphasized in this study, and budget constraints where each individual incurs a treatment cost. Ratio-form examples include risk-based constraints and certain fairness constraints such as conditional statistical parity. We emphasize that fairness metrics are diverse, and we do not claim coverage of all possible formulations. Appendix E.7 (lines 1397–1450) provides detailed demonstrations showing how MOSIC incorporates these representative constraints.
>
> Constraints outside these two families include those involving higher-order moments, such as enforcing SMD $<$ 0.1 or bounding on the variance of subgroup ATE, are not directly supported. These constraints do not reduce to linear or ratio expressions in the subgroup indicator.
>
> To accurately address the comment for finite-sample guarantees with constraints, we kindly ask for clarification. Our current Lemma 2 provides guarantees of approximate feasibility at the local min-max point given a fixed sample size.
>
> **Q2**
> Because our objective is non-convex and non-concave, we cannot guarantee convergence to an identical solution under different initializations. Instead, we evaluate stability empirically. For a fixed train–test split, we reran MOSIC-MLP six times with the same constraints but different random initializations and examined whether the resulting feature contributions remained consistent. As shown in Figures E.8.2, the SHAP value patterns are highly similar across runs, suggesting that the learned subgroups are stable with respect to initialization.

---

### Official Review · Reviewer_tNPX · 2025-10-29

**Soundness:** 3
**Presentation:** 3
**Contribution:** 2
**Rating:** 4
**Confidence:** 3

**Summary:**

This paper introduces MOSIC, a unified optimization framework for identifying optimal treatment subgroups under multiple practical constraints such as subgroup size and propensity overlap. Unlike conventional two-step CATE-based approaches, MOSIC formulates the task as a constrained min–max optimization and solves it using a modified Gradient Descent–Ascent algorithm. The authors establish theoretical guarantees for feasibility and local optimality and validate the approach on both synthetic and ICU datasets (eICU, MIMIC-IV). The framework is model-agnostic, compatible with various CATE estimators and model architectures, and can incorporate additional constraints such as fairness and safety.

**Strengths:**

- Introduces an optimization-based framework that integrates multiple constraints directly into subgroup identification, addressing the limitations of post-hoc filtering methods.
- Provides theoretical guarantees for convergence, feasibility, and approximate satisfaction of nonconvex constraints.
- Empirical evaluations show improvements in subgroup ATE, size control, and covariate balance on both synthetic and real-world datasets.
- Offers an open-source implementation with transparent algorithmic details, enhancing reproducibility.

**Weaknesses:**

- The method’s scalability in high-dimensional settings is not fully addressed, especially given the use of $\gamma$-GDA with many constraints.
- Limited exploration is provided on uncertainty quantification or confidence intervals for subgroup ATEs, which is critical for decision-making in clinical applications.
- Numerical stability concerns of $\gamma$-GDA are only discussed theoretically; empirical results showing training dynamics (e.g., convergence plots, constraint violation over iterations) would substantiate the stability claim.

**Questions:**

- The imbalance criterion should be SMD > 0.1 rather than 0.2 according to Austin (2009).
- The abbreviation “LR” in line 079 likely refers to Logistic Regression, which should be clearly defined when it first appears.
- What is the computational complexity and scalability of $\gamma$-GDA when applied to high-dimensional data or a large number of constraints?
- Can the authors discuss how subgroup interpretability is maintained when using black-box backbones such as MLPs and whether any post-hoc explainability tools were explored?

---

> ### Author Response · Authors · 2025-11-23
>
> We appreciate the reviewer's feedback. We have updated the abbreviation "LR" in line 080 to make it clear.
>
> **W1\&Q3**
> In terms of the data dimensionality, gradient-based methods, including GDA, are generally well-suited for high-dimensional optimization. The per-iteration cost of GDA grows linearly with the number of parameters because each update only requires computing gradients of the objective. This makes GDA more scalable than alternative approaches for subgroup identification, such as CAPITAL, which requires combinatorial search.
>
> As for the number of constraints, each constraint contributes one evaluation per update. The constraint families considered in our work (linear and ratio-form constraints) require at most $n$ multiplications, where $n$ is the number of samples. Thus, the per-iteration cost of computing the loss is $O(nm)$, where $m$ is the number of constraints. In our problem formulation, there are n overlap constraints, but each overlap constraints contributes only one multiplication, so the overall computation cost for overlap constraint is $O(n*1)= O(n)$ rather than $O(n^2)$. As a result, even though the number of constraints is large, the per-iteration cost remains small.
>
> We also provide the empirical runtime analysis in Appendix I (line 1711). Each method is run three times and average run time is reported; these experiments are run on CPU only to reflect clinical computing environments with limited resources. As shown, MOSIC's runtime at the optimization stage is trivial; thus, computing resources won't be a bottleneck for deployment.
>
> **W2**
> To quantify uncertainty, we computed the asymptotic 95\% confidence intervals based on the influence function of the AIPTW estimator.
> Figure E.1.1 (line 1215) reports the distribution of CI widths across 100 independent train–test splits, showing the anticipated increase in uncertainty as subgroup size decreases.
>
> **W3**
> We appreciate the reviewer's suggestion to show training dynamics to substantiate the stability claim. We have included the training curve in the appendix (Figure E.9.1, line 1585).
>
> **Q1**
> We agree that 0.1 is stricter and commonly used in epidemiology studies. However, Austin (2009) notes that "For modest sample sizes, one could expect standardized differences that exceed 0.20 (20 percent) even when the propensity-score model was correctly specified." Given that both of our real-world cohorts are relatively small (13361 patients in eICU and 6516 in MIMIC), we chose SMD $>$ 0.2 as the primary threshold to reduce the risk of flagging spurious imbalance driven by sample size limitations. This threshold is also frequently used in epidemiology studies. [1-2].
>
> We have rerun experiments and evaluated models using the stricter SMD $>$ 0.1 criterion (Appendix E.6). Although the absolute number of unbalanced covariates increases across all methods due to inherent data size limitations, the relative comparison remains the same: MOSIC achieves comparable balance while yielding higher subgroup ATEs (Figure E.6.1 at line 1388, and Figure 2 at line 400).
>
> On the MIMIC dataset, where dataset is considerably smaller than eICU, MOSIC appears to have a higher number of unbalanced features (Figure E.6.1(b)) under SMD$>0.1$, but most differences are not statistically significant (Table E.6.2, line 1422). DragonNet demonstrates significantly lower feature imbalance, but our method obtains significantly higher subgroup ATE improvement, as shown in Figure 2b (line 400) and Table D.0.4. (line 1164). Thus, this dataset-specific fluctuation does not alter the overall conclusion.
>
> **Q4**
> For black-box backbones, we conducted a post-hoc explainability analysis using SHAP values. This analysis highlights the features that contribute most strongly to subgroup membership (Figures E.8.1 - E8.2, line 1513-1558).
>
> We also compared the SHAP-derived feature importance patterns with those generated by DT model trained on the same data. The dominant features identified by SHAP are generally consistent with the rule structure revealed by the decision tree (line 512-514).
>
> Overall, MOSIC supports model backbones with different levels of interpretability. When a black-box model is used, we provide SHAP analysis to offer insight into subgroup membership drivers, though it does not yield explicit decision rules. We encourage practitioners to choose the backbone that best matches their interpretability needs.
>
> [1] Kang J, Ji E, Kim J, Bae H, Cho E, Kim ES, Shin MJ, Kim HB. Evaluation of patients’ adverse events during contact isolation for Vancomycin-Resistant Enterococci using a matched cohort study with propensity score. JAMA network open. 2022 Mar 1;5(3):e221865-.
>
> [2] Khalid SI, Massaad E, Shin JH. Assessing the prognostic impact of body composition phenotypes on surgical outcomes and survival in patients with spinal metastasis: a deep learning approach to preoperative CT analysis. Journal of Neurosurgery: Spine. 2024 Dec 20;1(aop):1-0.

---

> > ### Comment · Reviewer_tNPX · 2025-11-25
> >
> > Thank you for the rebuttal. The clarifications and new results address my main concerns. I am keeping my original score.

---

### Official Review · Reviewer_dfgj · 2025-11-01

**Soundness:** 2
**Presentation:** 3
**Contribution:** 2
**Rating:** 6
**Confidence:** 2

**Summary:**

The paper proposes MOSIC, a model-agnostic framework for identifying treatment-benefit subgroups under multiple constraints such as minimum size and overlap. It formulates a constrained optimization with ReLU-gated Lagrangian penalties and provides local min–max guarantees. Experiments on synthetic data and two ICU datasets show that MOSIC achieves higher subgroup treatment effects with fewer imbalance violations than prior subgroup discovery methods.

**Strengths:**

The paper proposes a clear optimization framework for subgroup identification under multiple constraints. The formulation is well defined and the paper is clearly written. The experiments are consistent across datasets and show some improvement in subgroup ATE and balance, though mainly within controlled and limited settings

**Weaknesses:**

1. The largest concern is the limited baselines: the paper includes CAPITAL, causal trees, causal forests, and outcome-weighted learning, but exclude stronger HTE estimators such as DR-learner, R-learner, BART, and locally centered causal forests. These are standard in healthcare and could change the ranking. Overlap-weighted estimators are also missing and align directly with the overlap constraint. It would also be helpful to discuss recent constrained policy methods from off-policy evaluation, safe RL, and constrained contextual bandits that optimize under budgets or risk

2. Experimental setup lacks robustness and diagnostic depth.
The evaluation relies on AIPTW without clear cross-fitting, which can bias subgroup ATEs when selection and estimation share nuisance models. The reliability metric is only the count of covariates with standardized mean difference>0.2 after IPTW, which is coarse and does not measure magnitude or joint imbalance. Overlap enforcement is only evaluated during training, not validated on test sets.

3. Causal validity in ICU cohorts is not well supported.
The treatment window (10 hours before to 24 hours after ICU admission) risks immortal time bias if early deaths are excluded. Confounding from indication severity is possible, especially for steroids, yet no sensitivity analysis or negative control outcomes are reported. The added safety constraint on GCS is useful, but without external validation or clinician input, it’s unclear whether the selected subgroups are clinically meaningful or plausible.

**Questions:**

1. How do you ensure unbiased AIPTW estimates if subgroup selection and nuisance estimation share data? Was any cross-fitting or sample splitting applied?
2. Can you evaluate overlap on test data, e.g., showing the distribution of estimated propensities or proportion outside [0.05, 0.95] within selected subgroups?
3. Why not compare against overlap-weighted baselines or modern CATE learners like DR-/R-learner or BART?
4. Could you report uncertainty for subgroup ATEs (e.g., influence-function or bootstrap CIs) and include a sensitivity analysis for unmeasured confounding in the ICU data?

---

> ### Author Response · Authors · 2025-11-23
>
> We thank the reviewer for the insightful comments. Below, we address your comments separately.
>
> **W1 \& Q3**
> We have added a discussion on recent constrained policy methods in the related work section (line 070-073) and included DR-learner, R-learner, BART, and overlap-weighted (OW) estimator in our evaluation, providing a more complete comparison against modern estimators.
>
> Fig. E.2.1 (line 1262) shows that the same conclusions remain unchanged: MOSIC consistently outperforms baselines with respect to subgroup ATEs and feature balance.
>
> We clarify that OW cannot serve directly as a subgroup-selection baseline because it outputs Average Treatment effect on the Overlap Population rather than sample-level CATEs needed for ranking or selecting subgroups.
> To compare against OW, we implemented a MOSIC variant (MOSIC-OW) that replaces the primary objective (line 170–172 in the paper) with overlap-weighted outcomes: $\phi(x_i,a_i,y_i) = [a_i*(1-e(x_i) ) + (1-a_i)*e(x_i)] * y_i, $.
>
> The poor performance of MOSIC-OW stems from a mismatch in estimands: optimizing under OW focuses on the overlap population within the subgroup, rather than the subgroup itself, which misaligns with the goal of subgroup identification.
>
> **W2 \& Q1**
> We clarify that all nuisances and the subgroup model are trained on the training set only. At evaluation time, we apply the subgroup model to the held-out test set and compute AIPTW using nuisances fitted on the training set. Because the test set is never used for either step, the reported AIPTW is not affected by data sharing in evaluation. Sharing data within the training set may lead to suboptimal performance on the test set, but this affects all methods equally.
>
> To assess whether internal sample-splitting helps, we implemented 5-fold cross-fitting on the training data: nuisances were trained on 4 folds and used to generate CATEs on the held-out fold, and the subgroup model was trained on these cross-fitted CATEs. Evaluation remained on the test set, where nuisances re-fit on the full training set. The results (Figure E.3.1, line 1277) closely match our original pipeline, suggesting that the benefit of decoupling nuisance and subgroup estimation is limited in our setting, likely because further splitting of a modest dataset weakens nuisance estimation.
>
> **W2 \& Q2**
> We have rerun experiments and now report overlap on the held-out test sets. Figure E.5.1. (line 1367) reports the proportions of samples with propensities outside [0.05,0.95] within the selected subgroups, and Figure 3 (line 443) reports the proportions outside the range enforced during training: [0.02,0.98] on synthetic data and [0.01,0.99] on real-world data. i.e., improving overlap relative to [0.05,0.95] is not enforced but is still reported for completeness. On real-world data, we relaxed the threshold to [0.01,0.99] because a substantial fraction of ICU patients naturally fall outside [0.05,0.95].
>
> When no overlap constraint is used ($\alpha=0$), the low-overlap proportion in the selected subgroup mirrors that of the full test set. When $\alpha>0$, these proportions drop substantially, showing that MOSIC’s overlap constraint improves overlap during both training and evaluation.
>
> **W3 \& Q4**
> We computed the 95\% CI based on the influence function of the AIPTW estimator.
> Figure E.1.1 (line 1215) reports the distribution of CI widths across independent train–test splits, showing the anticipated increase in uncertainty as subgroup size decreases. For unmeasured confounding, we report E-values for the subgroup ATEs. Figure E.1.2 (line 1232) shows that the E-values range from 1.2 to 1.35 across group sizes, indicating that a modest unmeasured confounding could explain away the estimated ATE. This aligns with the corresponding risk ratios on the right ($\approx$ 0.93-0.97), which reflect small but directionally stable protective effects.
>
> While such analyses show the causal strength and uncertainty of our estimates, we acknowledge that they did not fully address all concerns, such as immortal time bias. But we emphasize that these concerns pertain to the cohort-construction pipeline and nuisance estimation, and thus affect **all** baselines equally. Addressing these challenges requires domain-specific confounding control and additional sensitivity analysis, which are beyond the scope of the subgroup-selection problem in this study.
> Moreover, because MOSIC is agnostic to the CATE estimator, an improved confounder adjustment can be readily incorporated without modifying the optimization component.
> We have revised the manuscript to explicitly acknowledge these limitations (line 534-539).

---

### Author Response · Authors · 2025-12-01
**Summary of Reviewer Concerns and Rebuttal Responses**

We summarize the concerns raised by the reviewers and how they were addressed during rebuttal.

1. **Baselines and Related Work** (dfgj W1 & Q3):

    Reviewer dfgj requested stronger baselines (DR-learner, R-learner, BART, overlap-weighted estimators) and a discussion of recent constrained-policy methods.

    *Response*: We added all requested baselines and expanded the related-work discussion. The additional experiments confirm that MOSIC continues to outperform alternatives; the conclusions remain unchanged.

2. **Sample-splitting and AIPTW Evaluation** (dfgj W2 & Q1):

    Concern: AIPTW estimates might be biased if subgroup selection and nuisance estimation share data.

    *Response*: We clarified that all nuisance and subgroup models are trained solely on the training set; the test set is never used for either step. Therefore, AIPTW evaluation is not affected by data sharing. Data sharing may lead to suboptimal performance on the test set but affects all methods equally. Further, we conduct an additional experiment to see if sample-splitting on the training set can benefit the subgroup selection. Results show minimal benefit, likely due to weakened nuisance estimation when splitting relatively modest datasets.

3. **Overlap evaluation on test set** (dfgj W2 & Q2):

    Concern: Overlap enforcement was only evaluated during training.

    *Response*: We added the overlap evaluation on the held-out test sets as suggested. MOSIC effectively enforced the overlap constraint and substantially reduced the low-overlap proportion in the selected subgroup.

4. **Uncertainty Quantification** (dfgj W3 & Q4; tNPX W2):

    Both reviewers asked for confidence intervals, dfgj further asks for sensitivity analysis of unmeasured confounding.

    *Response*: We reported 95\% CIs and E-values for subgroup ATEs. These quantify uncertainty and sensitivity for unmeasured confounding. We acknowledge that some broader concerns (e.g., immortal-time bias) remain, but emphasize that these limitations affect all baselines equally and lie outside the scope of this study.

5. **Computational Complexity** (tNPX W1 & Q3; puYb W2 \& Q1 & Q3):

    Concern: time complexity with respect to the data dimension and the number of constraints, and real-world practicality.

    *Response*: We provided a theoretical per-iteration complexity analysis and empirical runtime measurements. MOSIC adds minimal overhead beyond nuisance estimation, and its runtime is substantially smaller than CAPITAL, the closest comparable method that also performs constrained subgroup selection. This addresses both scalability and the practicality concerns raised by reviewers.

6. **Training dynamics** (tNPX W3):

    Concern: numerical stability and convergence behavior.

    *Response*: We added training-curve visualizations showing stable optimization dynamics.

7. **Feature-Imbalance Threshold (SMD > 0.1 vs. 0.2)** (tNPX Q1)

    Reviewer tNPX suggested SMD >0.1 for feature imbalance evaluation. We clarify that we chose 0.2 because of the data size limitation, which is explicitly discussed in the paper tNPX referred to (Austin 2009). We nevertheless added experiments using SMD > 0.1; though absolute imbalance counts increase, the relative comparison across methods remains unchanged.

8. **Interpretability** (tNPX Q4; puYb W3 & Q3):

    In response to tNPX’s request for post-hoc interpretability of the black-box backbones, we added a SHAP analysis to illustrate feature contributions. puYb recommended providing a human-readable example of subgroup selection rules, so we moved the decision-rule example from the appendix into the main text for greater visibility.

9. **Narrative Structure** (puYb W1): puYb suggested a forward-design narrative in our method section. In response, we revised Section 4.2.

10. **Clarification of constraints** (puYb Q1):

    puYb questions about which constraints MOSIC can naturally support and whether finite-sample guarantees with constraints exist.

    *Response*: We clarified that MOSIC supports linear and ratio-form constraints with details in both the main text and Appendix E.7, and we further clarify constraints outside these forms (e.g., higher-order moments) are not supported. Regarding finite-sample guarantees with constraints, we explain our guarantee again, but we don't fully understand this question, so we asked for clarification.


11. **Stability Across Random Initializations** (puYb Q2):

    Concern: Whether MOSIC yields stable subgroups across different initializations.

    *Response*: We ran MOSIC-MLP six times with different initializations and analyzed SHAP patterns. Results show highly consistent feature attribution patterns, suggesting stable subgroup selection.

Overall, we provided additional experiments and clarifications to address all major concerns raised. These updates leave the core empirical conclusions unchanged and enhance the clarity and strength of the paper.

---

### Meta-Review · Area_Chair_DPC9 · 2026-01-06

**Summary:**

The paper addresses a relevant applied problem, but the conceptual contribution is limited. Reformulating a constrained subgroup selection problem as a differentiable min max objective and solving it via gradient descent ascent is a fairly standard technique in constrained optimization and does not, on its own, constitute a substantial methodological advance. The claimed novelty relative to existing policy learning, constrained optimization, and subgroup discovery methods is not clearly articulated, and it is unclear how the proposed framework fundamentally differs from Lagrangian based or bilevel formulations already studied in the causal inference and operations research literature.

On the theoretical side, the convergence guarantees are weak, establishing only local optimality and feasibility under conditions that are not especially informative for practice. The empirical results, while extensive, mainly demonstrate that directly enforcing constraints improves constraint satisfaction, which is expected by construction, and do not convincingly show superior decision quality compared to strong existing baselines. Overall, the paper does not provide sufficient theoretical depth or conceptual novelty to justify acceptance.

**Reviewer Scores:**

can't predict

---

### Decision · Program_Chairs · 2026-01-26

Reject